# Lateral interactions between protofilaments of the bacterial tubulin homolog FtsZ are essential for cell division

Fenghui Guan[1,2†], Jiayu Yu[3†], Jie Yu[1,2†], Yang Liu[3†], Ying Li[1†], Xin-Hua Feng[1,2], Kerwyn Casey Huang[4,5,6], Zengyi Chang[3*], Sheng Ye[1,2*]

[1]Innovation Center for Cell Signaling Network, Zhejiang University, Hangzhou, China; [2]Life Sciences Institute, Zheijiang University, Hangzhou, China; [3]The State Key Laboratory of Protein and Plant Gene Research, School of Life Sciences, Peking University, Beijing, China; [4]Department of Bioengineering, Stanford University, Stanford, United States; [5]Department of Microbiology and Immunology, Stanford University School of Medicine, Stanford, United States; [6]Chan Zuckerberg Biohub, San Francisco, United States

**Abstract** The prokaryotic tubulin homolog FtsZ polymerizes into protofilaments, which further assemble into higher-order structures at future division sites to form the Z-ring, a dynamic structure essential for bacterial cell division. The precise nature of interactions between FtsZ protofilaments that organize the Z-ring and their physiological significance remain enigmatic. In this study, we solved two crystallographic structures of a pair of FtsZ protofilaments, and demonstrated that they assemble in an antiparallel manner through the formation of two different inter-protofilament lateral interfaces. Our in vivo photocrosslinking studies confirmed that such lateral interactions occur in living cells, and disruption of the lateral interactions rendered cells unable to divide. The inherently weak lateral interactions enable FtsZ protofilaments to self-organize into a dynamic Z-ring. These results have fundamental implications for our understanding of bacterial cell division and for developing antibiotics that target this key process.
DOI: https://doi.org/10.7554/eLife.35578.001

**\*For correspondence:**
changzy@pku.edu.cn (ZC);
sye@zju.edu.cn (SY)

[†]These authors contributed equally to this work

**Competing interests:** The authors declare that no competing interests exist.

## Introduction

Bacterial cytokinesis is initiated by the formation of a ring-like structure termed the Z-ring, a polymeric assembly of the essential tubulin homolog FtsZ at future division sites (*Bi and Lutkenhaus, 1991*). Once formed, the Z-ring serves as a scaffold to recruit other cell division proteins that collectively constitute the divisome (*Dajkovic and Lutkenhaus, 2006*). During cell division, the Z-ring constricts at the leading edge of the invaginating septum, eventually causing a mother cell to divide into two daughter cells (*Bi and Lutkenhaus, 1991*).

FtsZ subunits have been suggested to interact through two putative sets of interfaces, longitudinal interfaces that join the subunits in a head-to-tail manner thereby forming a protofilament, and lateral interfaces that occur between protofilaments. FtsZ subunits readily assemble into protofilaments in vitro (*Mukherjee and Lutkenhaus, 1994*; *Romberg et al., 2001*), and crystal structures of FtsZ protofilaments have been determined for both straight (*Matsui et al., 2012*; *Tan et al., 2012*) and curved conformations (*Li et al., 2013*). In vitro, FtsZ protofilaments have been observed to further associate via lateral interfaces to form higher-order structures such as sheets (*Bramhill and Thompson, 1994*; *Erickson et al., 1996*; *González et al., 2003*; *Löwe and Amos, 1999*; *Löwe and*

**eLife digest** New cells form when existing cells divide. When a cell divides it narrows at one point, which eventually allows it to split in two. This basic process of division happens in cells from all species, although they do not all use the same mechanisms to achieve it. In bacteria, a structure called the Z-ring guides where the cell narrows and divides. Although the importance of the Z-ring in bacterial cell division is clear, how it works was not known.

A first step to understanding how the Z-ring works is to find out how it is made. The Z-ring consists of long 'protofilaments' made up of many copies of a protein called FtsZ. To find out how the protofilaments interact with each other to form the Z-rings, Guan, Yu, Yu, Liu, Li et al. studied the interactions between the FtsZ proteins in living cells. This revealed two key points of contact that allow two protofilaments to link together while aligned in opposite directions.

Further experiments in living cells showed that disrupting either contact point prevents the cells from growing correctly and can cause cells to die. Guan et al. also show that these contacts are weak, so two protofilaments can only link together when many of their FtsZ proteins interact.

Future research into how the Z-ring works can build upon these details of how the protofilaments interact. Because animal cells do not contain Z-rings, this could ultimately help researchers to design new antibiotics that can kill bacteria without affecting other cells.

DOI: https://doi.org/10.7554/eLife.35578.002

*Amos, 2000*; *Oliva et al., 2003*; *Yu and Margolin, 1997*). While several studies strongly suggested that lateral interfaces across protofilaments are important for FtsZ function (*Dajkovic et al., 2008*; *Lan et al., 2009*; *Milam et al., 2012*; *Szwedziak et al., 2014*), the precise nature and the functional relevance of these lateral interfaces remain largely unclear.

Although bacterial cell division has been actively investigated for decades, the in vivo nanoscale organization of the Z-ring has not been well defined thus far. Conventional fluorescence microscopy depicts the Z-ring as a smooth, closed ring, with individual protofilaments not resolvable (*Pogliano et al., 1997*; *Sun and Margolin, 1998*). Based on in vitro assembly studies (*Erickson et al., 1996*; *Löwe and Amos, 1999*; *Löwe and Amos, 2000*; *Mukherjee and Lutkenhaus, 1994*), the Z-ring was initially modeled as a few single continuous polymers that wrap around the cell. Later, electron cryotomography suggested that the Z-ring is composed of individual FtsZ protofilaments that do not obviously interact laterally, scattered in a narrow band around the circumference of the cell (*Li et al., 2007*). Super-resolution light microscopy indicated that FtsZ protofilaments form randomly oriented, multi-layered, discontinuous clusters within the Z-ring (*Biteen et al., 2012*; *Buss et al., 2013*; *Coltharp et al., 2016*; *Fu et al., 2010*; *Holden et al., 2014*; *Jacq et al., 2015*; *Rowlett and Margolin, 2014*; *Si et al., 2013*; *Strauss et al., 2012*). By contrast, recent electron cryotomography studies found a small, single-layered band of FtsZ protofilaments parallel to the membrane (*Szwedziak et al., 2014*), and showed that a complete ring of FtsZ is not required to initiate constriction in the early stages of cytokinesis (*Yao et al., 2017*). The link between this diverse set of conformations and Z-ring dynamics is challenging to parse without structural knowledge of the full suite of inter-subunit interactions.

To address the nature and in vivo role of FtsZ lateral interactions, we solved the structure of *Mycobacterium tuberculosis* FtsZ (MtbFtsZ) in a double-stranded protofilament state. Comparison of this structure with that of MtbFtsZ in a different double-stranded protofilament state that we previously determined (*Li et al., 2013*) revealed two different inter-protofilament lateral interfaces. Using a combination of site-directed mutagenesis and phtotocrosslinking studies, we demonstrate that these lateral interfaces occur in living cells, and are critical for mediating cell division through the assembly of protofilaments into a functional Z-ring.

## Results

### Structural analysis reveals lateral interfaces for FtsZ protofilament bundling

FtsZ proteins from phylogenetically divergent species are known to assemble into polymers with multiple morphologies in a nucleotide-dependent manner (*Erickson et al., 1996*; *Löwe and Amos, 1999*; *Löwe and Amos, 2000*; *Lu et al., 1998*; *Oliva et al., 2003*; *Popp et al., 2010*; *White et al., 2000*). Our electron microscopy analysis showed that MtbFtsZ and FtsZ from *Escherichia coli* (EcFtsZ) are able to form protofilament bundles in vitro in the presence of DEAE-dextran (*Figure 1A,B*). The fact that protofilaments of both EcFtsZ and MtbFtsZ are able to form such assemblies, as observed previously (*Erickson et al., 1996*; *Löwe and Amos, 1999*), suggests that the lateral interface of FtsZ protofilaments is a common and conserved characteristic.

FtsZ subunits were previously observed to assemble into single- and double-stranded filaments at physiological concentrations (*Chen et al., 2007*; *Oliva et al., 2003*; *White et al., 2000*). Our previous structural analysis of MtbFtsZ also revealed the formation of double-stranded and curved filaments, arranged in an antiparallel fashion (*Li et al., 2013*). From the MtbFtsZ structure (*Li et al., 2013*), we observed an inter-protofilament interface located on the external faces of strands S7 and S10 in the C-terminal subdomain (lateral interface 1, *Figure 1C*) (*Li et al., 2013*). However, the existence of only a single lateral interface within such an antiparallel arrangement of protofilaments would be self-limiting and lead only to the formation of double-stranded filaments. Formation of bundles composed of more than two FtsZ protofilaments requires additional lateral interfaces between the opposite sides of the protofilaments. We have now identified candidates for these interfaces in a new hexagonal crystal of MtbFtsZ, which has been determined to an $R_{free}$ factor of 27.3% at a resolution of 2.7 Å (Materials and methods, *Table 1*). Compared with our earlier MtbFtsZ structure (*Li et al., 2013*), our newly determined MtbFtsZ structure is similarly double-stranded and reveals curved filaments in an antiparallel arrangement. However, in this new structure, the inter-protofilament interface is located at the external faces of helices H3, H4, and H5 in the N-terminal subdomain (lateral interface 2, *Figure 1D*).

In the previously identified lateral interface (*Li et al., 2013*), Arg229 of one subunit and Asp301 of the other formed two pairs of salt bridges, burying a surface area of approximately 210 Å$^2$ (*Figure 1E*). By contrast, the lateral interface in the new MtbFtsZ structure is composed of basic residues Arg76, Lys77, Lys83, Arg119, and Lys120 and acidic residues Glu80, Glu87, and Glu153 from both interacting subunits, burying a larger surface area of ~870 Å$^2$ (*Figure 1D*). These residues form charged complementary surfaces, suggesting the existence of electrostatic interactions. The charged residues involved in both lateral interfaces are generally conserved (*Figure 1—figure supplement 1*), indicating that they are functionally relevant. Such an electrostatic nature was predicted in earlier studies probing the effects of pH and ionic strength on FtsZ protofilament bundling (*Beuria et al., 2006*). Interestingly, in the previous MtbFtsZ structure (*Li et al., 2013*), only two of the three FtsZ subunits (A and B) in each protofilament participated in such lateral interactions, whereas the charged residues Arg229 and Asp301 in subunit C were ~6 Å apart (*Figure 1E*). These in vitro observations suggest the presence of weak lateral interactions between MtbFtsZ protofilaments, in agreement with earlier electron microscopy studies as well as predictions based on kinetic modeling (*Lan et al., 2008*).

Guided by the similarities in amino acid sequence and tertiary structure between MtbFtsZ and *Staphylococcus aureus* FtsZ (SaFtsZ), as well as the two lateral interfaces we have identified in MtbFtsZ filaments, we attempted to construct a model for sheet-like structures of FtsZ filaments. In light of the two MtbFtsZ structures, we initially constructed two different MtbFtsZ lateral dimer structures. Each subunit in these dimeric structures was subsequently superimposed on the SaFtsZ subunit in an SaFtsZ protofilament (*Matsui et al., 2012*) by aligning their main-chain atoms to generate a hybrid filament in which an MtbFtsZ protofilament pairs with an SaFtsZ protofilament. The MtbFtsZ structure in such a hybrid filament was then replaced with the SaFtsZ structure to generate an SaFtsZ filament. The final model contains four SaFtsZ protofilaments that associate laterally to form an antiparallel sheet-like structure (*Figure 1F*). This structure is very similar to that observed for EcFtsZ (*Erickson et al., 1996*) and *Methanococcus jannaschii* FtsZ (*Löwe and Amos, 1999*), suggesting that the lateral interfaces observed by X-ray crystallography are identical to those observed by electron microscopy.

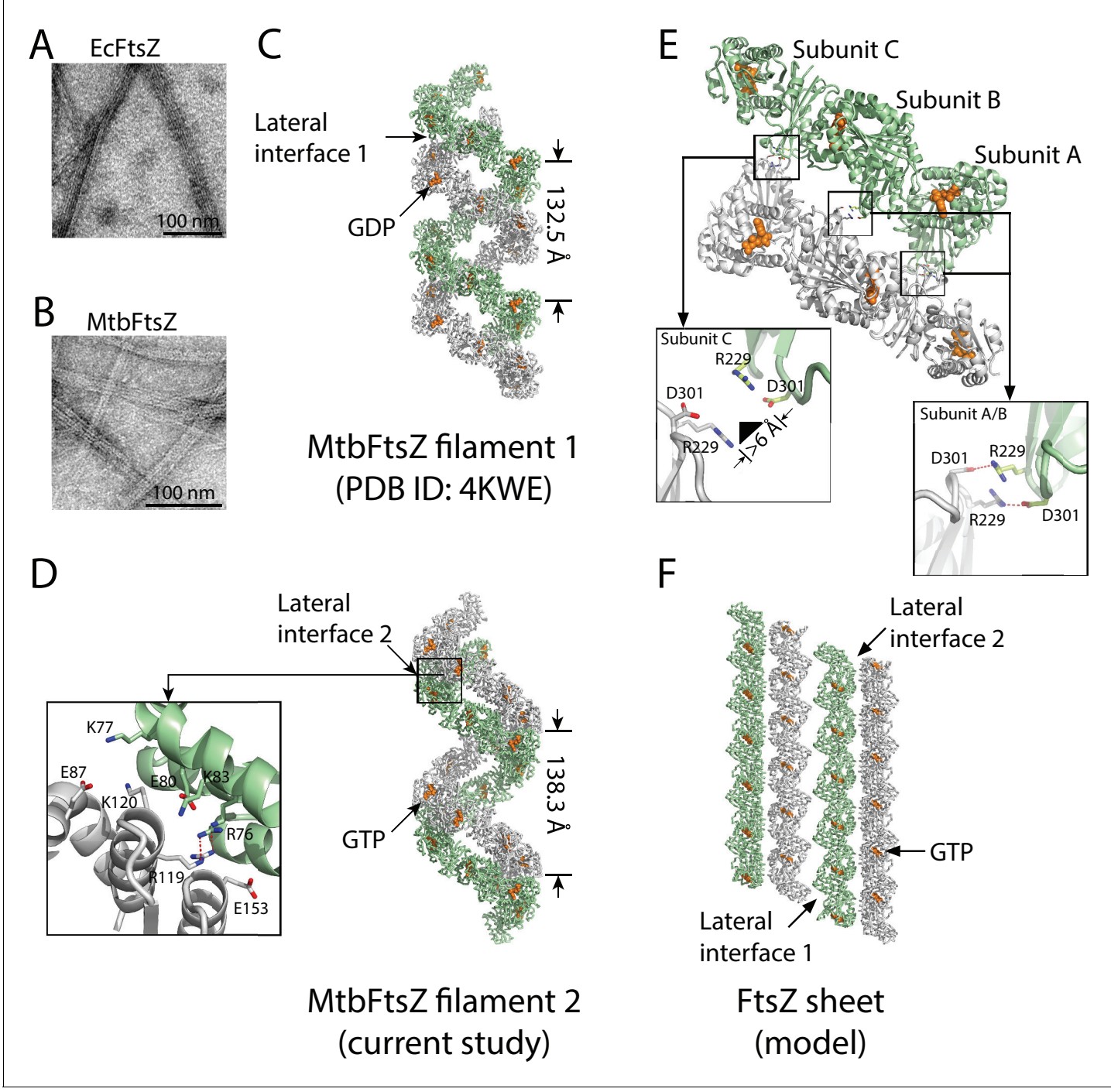

**Figure 1.** Structures of double-stranded MtbFtsZ-GDP and MtbFtsZ -GTP protofilaments reveal lateral contacts across FtsZ protofilaments. (**A, B**) Electron micrographs of protofilament bundles of EcFtsZ-GTP (**A**) and MtbFtsZ-GTP (**B**). Both were polymerized with the addition of 0.6 mg/mL DEAE-Dextran, and in the presence of 2 mM GTP. (**C, D**) Cartoon representations of double-stranded MtbFtsZ-GDP (**C**; PDB ID: 4KWE) and MtbFtsZ-GTP (**D**; this study) protofilaments containing a total of 24 subunits. The helices have a pitch of 132.5 Å for MtbFtsZ-GDP (**C**) and 138.3 Å for MtbFtsZ-GTP (**D**) protofilaments. Each structure reveals unique lateral interactions across the protofilaments. Inset: atomic details of the lateral interface of the double-stranded MtbFtsZ-GTP protofilaments. (**E**) Molecular details of the lateral interface of the double-stranded MtbFtsZ-GDP protofilaments shown in (**C**). Inset: atomic details of the lateral interactions. (**F**) A structural model for sheet-like bundles of FtsZ protofilaments. Ribbon representation of four straight FtsZ-GTP protofilaments (each containing six subunits, arranged in an antiparallel fashion).

DOI: https://doi.org/10.7554/eLife.35578.003

The following figure supplement is available for figure 1:

*Figure 1 continued on next page*

*Figure 1 continued*

**Figure supplement 1.** Multiple sequence alignment of FtsZ and secondary structure elements.

DOI: https://doi.org/10.7554/eLife.35578.004

## FtsZ filaments interact at lateral interfaces in living cells

To probe inter-protofilament contacts in living cells, we utilized an in vivo photocrosslinking approach in which we replaced each of the corresponding interfacial amino acid residues with p-benzoyl-L-phenylalanine (pBpa), an unnatural photoactive amino acid that, upon UV irradiation, forms a biradical that can abstract an H atom from C-H bonds at a distance of ~3–4 Å to form a covalent adduct (*Chin et al., 2002*; *Chin and Schultz, 2002*; *Fu et al., 2013*; *Sato et al., 2011*; *Zhang et al., 2011*). Plasmids carrying mutated *ftsZ* genes were first transformed into an *ftsZ* conditional-null strain LY928-Δ*ftsZ*, whose genome contains the gene encoding the orthogonal aminoacyl-tRNA synthetase and tRNA needed for the incorporation of pBpa (*Wang et al., 2016*). Photocrosslinking analyses were then performed for the FtsZ-pBpa variants that were able to rescue cell growth (*Figure 2A* and *Figure 4—figure supplement 1*, Materials and methods). Upon irradiation with long-wavelength UV light, we found that FtsZ-pBpa variants R78pBpa, N79pBpa, D82pBpa, R85pBpa, R89pBpa, K155pBpa, and S231pBpa produced covalently linked homodimers, as demonstrated by immunoblotting analysis (*Figure 2B*). The same set of pBpa variants of FtsZ were expressed in an *E. coli* strain that also expresses the AviTagged form of wild type FtsZ, and the putative photocrosslinked dimers were then probed with either an anti-FtsZ antibody (which recognizes both the pBpa variant and the AviTagged wild-type FtsZ forms) or with a streptavidin-alkaline phosphatase conjugate (which only recognizes the AviTagged wild-type FtsZ). When probing with the anti-FtsZ antibody, doublet bands reflecting the migration positions of both the FtsZ monomer and dimer were detected (*Figure 2B*). By contrast, when probing with the streptavidin conjugate, only single bands at both the monomer and the dimer positions (corresponding to the higher molecular weight band in the anti-FtsZ immunoblot) were detected. These photocrosslinking results clearly demonstrate that both lateral interfaces mediate interactions between FtsZ subunits in living cells.

To obtain unbiased confirmation of the presence in living cells of the two crystallographically observed lateral interfaces, we further designed a random screening strategy (*Figure 3A*) (*Chin et al., 2002*; *Daggett et al., 2009*; *Liu and Schultz, 2010*; *Ryu and Schultz, 2006*; *Stricker and Erickson, 2003*). Instead of rationally introducing the unnatural amino acid pBpa via site-directed mutagenesis (*Figure 2A*), we randomly introduced it into the EcFtsZ protein by generating a plasmid-borne library such that an in-frame TAG amber codon, which will be read as pBpa, was randomly inserted throughout the *ftsZ* gene (*Daggett et al., 2009*). This library was then

**Table 1.** X-ray data and refinement statistics.

| Data set | |
| --- | --- |
| Space group | P6$_5$22 |
| Unit cell | a = 100.5 Å, c = 138.3 Å |
| Resolution (Å) | 2.7 |
| Measured reflections | 167,791 |
| Unique reflections | 11,805 |
| Redundancy | 14.2 |
| Completeness (%, highest shell) | 99.3 (99.5) |
| Mean I/σI (highest shell) | 44.1 (1.6) |
| Rsym (%, highest shell) | 10.7 (100) |
| **Refinement** | |
| Resolution (Å) | 2.7 |
| Number of reflections |F| > 0 σF | 10,593 |
| R-factor/R-free (%) | 21.1/25.3 |
| Number of protein atoms | 2203 |
| Number of GTP molecules | 1 |
| Number of water molecules | 0 |
| rmsd bond lengths (Å) | 0.008 |
| rmsd bond angles (°) | 0.97 |

DOI: https://doi.org/10.7554/eLife.35578.005

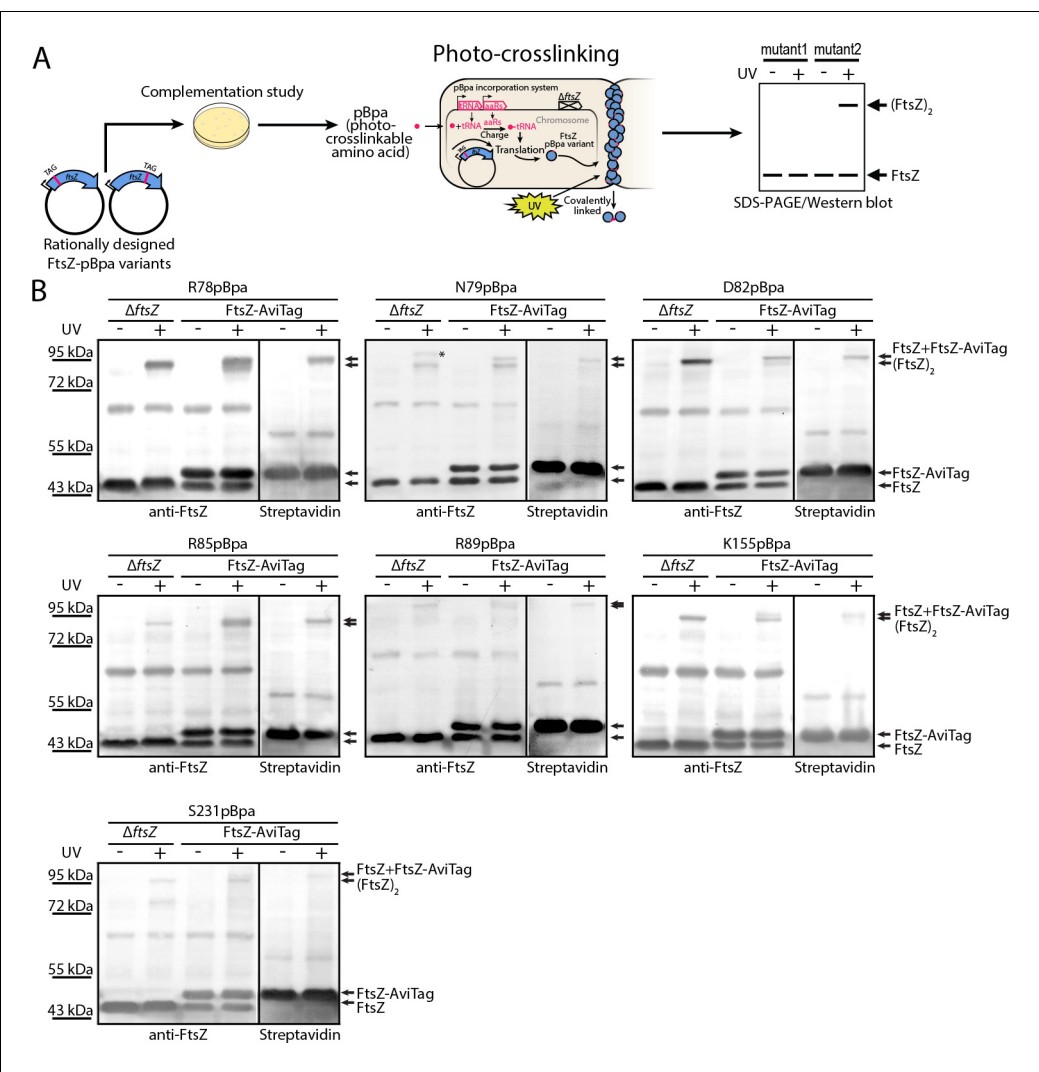

**Figure 2.** In vivo photocrosslinking characterization of EcFtsZ variants in which pBpa was introduced at specific sites validates inter-protofilament lateral interactions. (**A**) Schematic illustrating the rationale of in vivo photocrosslinking analysis via incorporation of the unnatural amino acid pBpa at specific residue positions of FtsZ. (**B**) Results of blotting analyses of photocrosslinked products with lysates of *E. coli* cells (LY928-Δ*ftsZ* or LY928-FtsZ-AviTag) expressing the indicated pBpa variant of FtsZ, using antibodies against EcFtsZ (left) or using alkaline phosphatase-conjugated streptavidin (right). Indicated on the right of each gel are the positions of the non-photocrosslinked monomers and the photocrosslinked dimers of the pBpa variant of FtsZ with or without the AviTag. The asterisk indicates the position of a crosslinked product of FtsZ and another as yet unidentified protein.
DOI: https://doi.org/10.7554/eLife.35578.006

transmformed into the LY928-Δ*ftsZ* strain to screen for variants that complemented the *ftsZ* conditional-null phenotype. These variants were then subjected to in vivo photocrosslinking analysis to identify pBpa variants of FtsZ that can form crosslinked dimers. We obtained 31 colonies that yielded crosslinked FtsZ products. We then sequenced the *ftsZ* genes from these 31 colonies and identified FtsZ-pBpa variants resulting from insertion of the TAG amber codon at 10 distinct sites. Our immunoblotting analysis indicated that photocrosslinked FtsZ dimers were formed for four of these ten variants (corresponding to pBpa incorporated at residue positions R78, D82, R85, or K140; *Figure 3B*). Among these four positions, residue K140 is located at the protofilament longitudinal interface and the FtsZ^K140A mutant was earlier demonstrated to complement an *ftsZ* conditional-null strain (*Li et al., 2013*; *Matsui et al., 2012*), while R78, D82, and R85 are located at lateral interface two observed in our crystal structure. As a negative control, we sequenced the *ftsZ* genes isolated

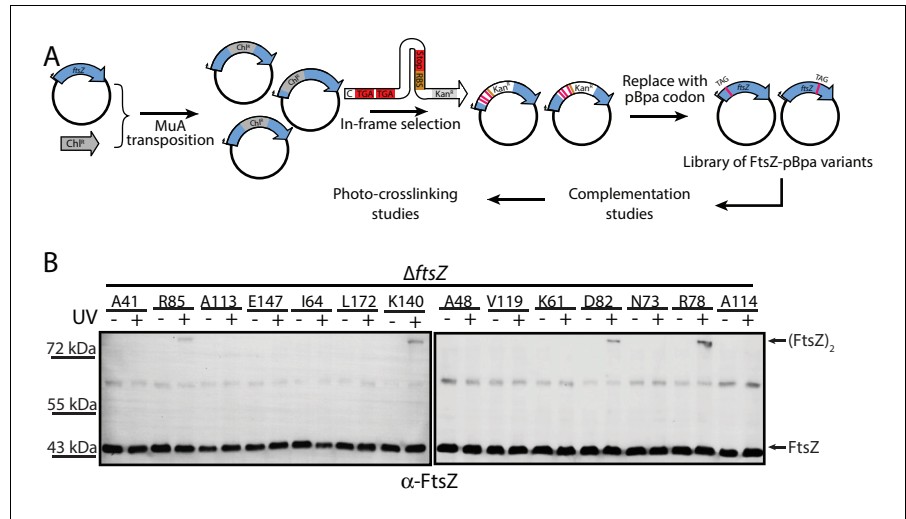

**Figure 3.** In vivo photocrosslinking characterization of randomly generated EcFtsZ pBpa variants. (**A**) Schematic illustrating the random in vivo photocrosslinking screening strategy for unbiased identification of amino acid positions mediating subunit interactions. (**B**) Immunoblotting analysis of the photocrosslinked products in lysates of *E. coli* LY928-Δ*ftsZ* cells that were transformed with the library of plasmids within which the in-frame TAG amber codon was randomly inserted throughout the *ftsZ* gene, as probed with antibodies against EcFtsZ.
DOI: https://doi.org/10.7554/eLife.35578.007

from 42 complementing colonies that did not generate any detectable photocrosslinked FtsZ products, from which we identified 12 distinct FtsZ-pBpa variants (corresponding to pBpa incorporated at residue positions G22, V37, A41, A48, K61, I64, N73, A113, A114, V119, E147, or L172). As expected, none of these 12 residues is located at either the longitudinal or lateral interfaces. Taken together, our photo-crosslinking analyses based on unbiased, random introduction of pBpa confirm the presence of at least two interfaces that are involved in FtsZ assembly in living cells, both consistent with our in vitro crystallographic analyses.

## The identified lateral interfaces between FtsZ filaments are physiologically relevant

Our photo-crosslinking analyses were performed for pBpa variants that could complement wild-type FtsZ. We were surprised to find that three variants (K121pBpa and D122pBpa from lateral interface 2, and D304pBpa from lateral interface 1) failed to complement (*Figure 4—figure supplement 1*). To exclude potential artifacts introduced by pBpa, we replaced each of the corresponding interfacial residues with hydrophobic leucine and then characterized these mutant proteins using a similar complementation approach (*Figure 4*, *Table 2*) (*Stricker and Erickson, 2003*). As with pBpa replacement, K121L and D304L failed to complement (*Figure 4*, *Table 2*). However, unlike D122pBpa, the D122L mutation was sufficient for complementation. This contrasting result with D122pBpa might be due to the bulkier size of the benzophenone-moiety side chain of pBpa compared to that of leucine.

Replacement of an interfacial hydrophilic residue (K or D) with the hydrophobic leucine could disrupt inter-protofilament interaction, or could induce protein misfolding. However, purified FtsZ$^{K121L}$ and FtsZ$^{D304L}$ retained similar GTPase activity to that of wildtype (data not shown), and assembled into protofilaments in a GTP-dependent manner, arguing against the possibility of protein misfolding. We performed photocrosslinking studies on the non-functional pBpa variants by expressing them in cells that also expressed the AviTagged wild-type FtsZ. Unlike functional pBpa variants (*Figure 2*), none of the three variants K121pBpa, D122pBpa, and D304pBpa produced any crosslinked dimer (*Figure 4—figure supplement 2*). These results indicate that the loss of FtsZ function in these variants is likely linked to a disruption of lateral interactions. Nevertheless, the dramatically distinct complementation results of the disruptive mutations of Ser231 and Asp304, two residues likely involved in direct interactions at lateral interface 1, raise an obvious concern as to whether Asp304 is important for other functions. To address this possibility, we generated two double mutants across

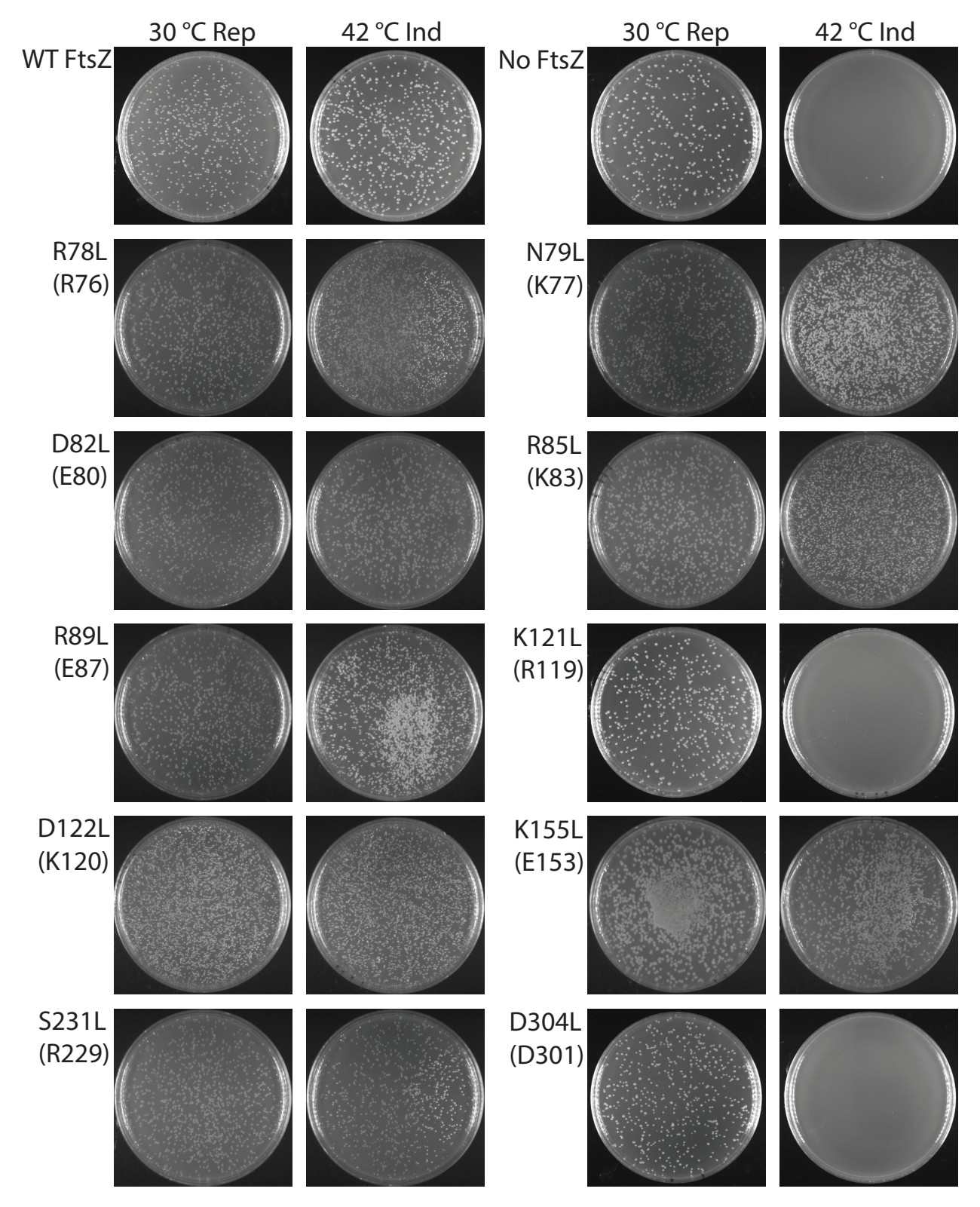

**Figure 4.** Complementation characterization of EcFtsZ mutants at inter-protofilament lateral interfaces. Ten EcFtsZ mutants were selected based on our crystallographic observations. Mutations were introduced by altering hydrophilic residues to hydrophobic leucine. The division phenotype was characterized using a Δ*ftsZ* strain expressing the EcFtsZ mutants, with conditional expression of wild-type FtsZ from a plasmid at 30°C but not 42°C. 'Rep' and 'Ind' indicate repression and induction media, respectively. For each mutant, the complementation assay was repeated three times.

*Figure 4 continued on next page*

*Figure 4 continued*

DOI: https://doi.org/10.7554/eLife.35578.008

The following figure supplements are available for figure 4:

**Figure supplement 1.** Complementation characterization of pBpa-incorporated EcFtsZ variants.

DOI: https://doi.org/10.7554/eLife.35578.009

**Figure supplement 2.** Immunoblotting analysis of three pBpa variants of EcFtsZ that failed to complement the *ftsZ* conditional-null strain shows absence of crosslinked dimers.

DOI: https://doi.org/10.7554/eLife.35578.010

the interface (D304L/S231E and D304L/S231Q) and observed complementation (*Figure 5*), demonstrating the formation of lateral interface one in vivo. Taken together, these data suggest that the two lateral interfaces we observed in vitro are important for FtsZ function in vivo, and lack of complementation is likely due to loss of lateral contacts.

## Weak lateral interactions exist between protofilaments

We initially postulated from the electrostatic complementarity along both lateral interfaces that short-range electrostatic interaction is the main driving force for lateral interactions. However, three lines of evidence led us to revisit this interaction mechanism. First, complementation results of presumably disruptive mutants on the lateral interface were less predictable than those of disruptive mutants on the longitudinal interface (*Li et al., 2013*). Second, residues on the lateral interfaces are either polar or electrostatic, and are only generally conserved. For example, the Arg229-Asp301 pair observed in MtbFtsZ becomes Ser231-Asp304 in EcFtsZ. Third, the two complementing double mutants across the lateral interface (D304L/S231E and D304L/S231Q) indicate that S231E or S231Q forms favorable interactions that compensate for the disruptive effect of D304L. We further mutated

**Table 2.** Complementation effects of inter-protofilament interface mutants.

| | Corresponding amino acid in MtbFtsZ | Complementation on plates | Complementation in liquid culture |
|---|---|---|---|
| Wildtype | | Yes | Yes |
| R78L | R76 | Yes | Yes |
| N79L | K77 | Yes | Yes |
| D82L | E80 | Yes | Yes |
| R85L | K83 | Yes | Yes |
| R89L | E87 | Yes | Yes |
| K121L | R119 | No | No |
| D122L | K120 | Yes | Yes |
| K155L | E153 | Yes | Yes |
| S231L | R229 | Yes | Yes |
| D304L | D301 | No | No |
| N78pBpa | | Yes | Yes |
| N79pBpa | | Yes | Yes |
| D82pBpa | | Yes | Yes |
| R85pBpa | | Yes | Yes |
| R89pBpa | | Yes | Yes |
| K121pBpa | | No | No |
| D122pBpa | | No | No |
| K155pBpa | | Yes | Yes |
| S231pBpa | | Yes | Yes |
| D304pBpa | | No | No |

DOI: https://doi.org/10.7554/eLife.35578.011

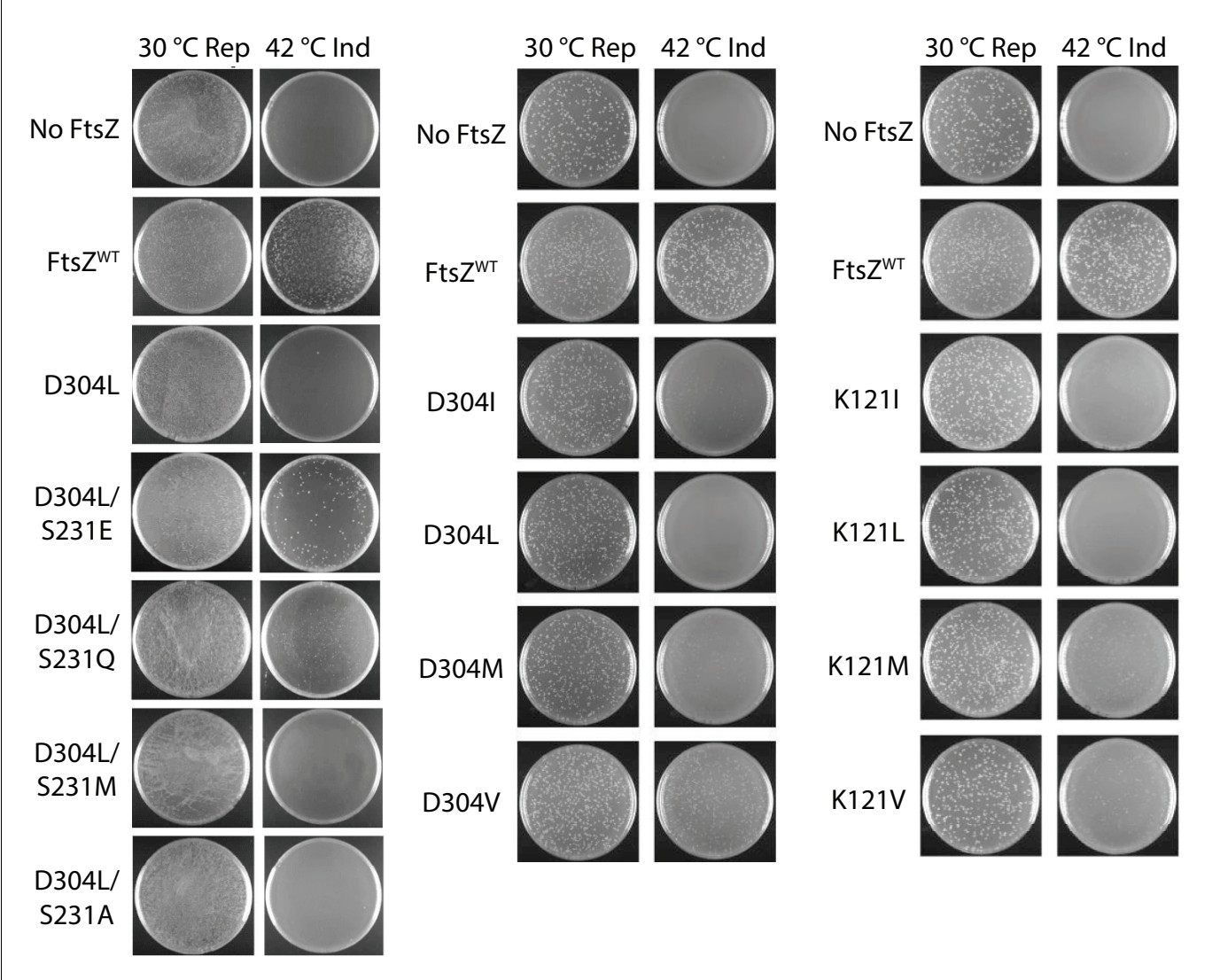

**Figure 5.** Complementation characterization of EcFtsZ mutants. Mutations were introduced by altering D304 and K121 to different hydrophobic residues. In addition to the D304L mutation, S231 was replaced several other amino acids. The division phenotype was characterized using a ΔftsZ strain with expression of the EcFtsZ mutants, and conditional expression of wild-type FtsZ from a plasmid at 30℃ but not 42℃. 'Rep' and 'Ind' indicate repression and induction media, respectively. For each mutant, the complementation assay was repeated three times.

DOI: https://doi.org/10.7554/eLife.35578.012

Asp304 to different hydrophobic residues and observed highly variable results; for example, D304V was able to complement (*Figure 5*). Mutagenesis of Lys121 revealed similar variability, with K121M and K121V able to complement (*Figure 5*). These results, together with those from double mutagenesis (*Figure 5*), suggest that lateral interactions are predominantly mediated by van der Waals interactions, which are sensitive to surface geometry; the charge complementarity may enhance these associations. Moreover, these results also suggest that lateral interactions between FtsZ protofilaments are much weaker on a per subunit basis in comparison with hydrophobic longitudinal interactions.

The free energy of protein-protein association is a balance between the intrinsic bond energy and the subunit entropy. The former favors association, while the latter disfavors association due to immobilizing a subunit (*Erickson, 1989*). This balance prompted us to examine whether protofilament formation is a prerequisite for lateral interactions to occur. To this end, we introduced A181E, a mutation known to disrupt the longitudinal interface (*Li et al., 2013*), into a set of pBpa variants of

FtsZ, including R78pBpa, N79pBpa, D82pBpa, R85pBpa, R89pBpa, K155pBpa, and S231pBpa, all of which produced covalently linked FtsZ dimers upon UV irradiation (*Figures 2B* and *3B*). We then performed in vivo photocrosslinking analysis with this set of A181E-containing pBpa variants, and found that photocrosslinked dimers were no longer detectable for all such variants (*Figure 6*). Thus, protofilament preassembly is required for lateral interactions to occur, consistent with the hypothesis that the lateral interactions are generally weak on a per-subunit basis. Nevertheless, the combined strength of all lateral interactions is presumably significant given that many interfaces are present along the protofilaments.

## Lateral interfaces are involved in Z-ring assembly

FtsZ subunits readily assemble into protofilaments in vitro (*Mukherjee and Lutkenhaus, 1994*; *Romberg et al., 2001*). Given that the intracellular concentration (~5.6 μM) (*Li et al., 2014*) is much higher than the critical concentration (~1 μM) (*González et al., 2003*), it is reasonable to assume that most FtsZ molecules assemble into protofilaments in vivo. Since our complementation studies revealed the importance of both lateral interfaces for FtsZ function, we next investigated whether FtsZ mutant proteins defective in lateral interactions can integrate into the Z-ring in living *E. coli* cells that also express wild-type FtsZ.

We first confirmed that these FtsZ mutant proteins (D304L and K121L) are still capable of forming GTP-dependent protofilaments (*Figure 7A*). This capacity indicates that, when the laterally disruptive FtsZ is co-expressed with wild-type FtsZ, they can stochastically copolymerize to form hybrid protofilaments. We assume that the fraction of laterally disruptive subunits incorporated into protofilaments follows a Binomial distribution with a mean corresponding to the cellular proportion of laterally disruptive FtsZ (*Figure 7—figure supplement 1*), and that this fraction will determine the number of effective lateral bonds that could form between protofilaments. Given that the lateral interactions are weak, we expect that there exists a critical fraction of laterally disruptive subunits within a protofilament, above which the combined lateral interactions are insufficient to exceed the entropic cost of immobilizing the protofilament. In this case, we expect a dramatic reduction in the probability of such a protofilament interacting with other protofilaments to incorporate into the Z-ring. Thus, when co-expressed with wild type FtsZ, if the cellular proportion of laterally disruptive FtsZs is low, most protofilaments will tolerate the small degree of lateral disruption and incorporate into the Z-ring, whereas a high proportion of laterally disruptive FtsZ will interfere with Z-ring formation. Our complementation studies have already suggested that without wild-type FtsZ, laterally disruptive FtsZ mutants are lethal. For a pool of intermediate size, the protofilaments whose fraction of laterally disruptive subunits is above the threshold will be excluded from the Z-ring, leaving those hybrid protofilaments with small fraction of mutant subunits to form a functional Z-ring. As a consequence, when the cellular proportion of laterally disruptive FtsZ increases, the fraction of such subunits in the Z-ring decreases (*Figure 7—figure supplement 2*).

This aforementioned rationale prompted us to co-express wild type FtsZ and fluorescent protein-fused mutant FtsZ, and use the midcell fluorescence signal as a proxy for Z-ring incorporation. We observed that the laterally disruptive mutants K121L and D304L and the laterally nondisruptive mutant R78L were all efficiently incorporated into the Z-ring (*Figure 7C*), when the cellular proportions of mutant FtsZ proteins were ~40% (*Figure 7B*). We then sought to increase the ratio of mutant FtsZ to wild-type FtsZ by using a stronger promoter to express mNeonGreen-tagged FtsZ variants. We introduced an amber codon between EcFtsZ and mNeonGreen for each variant to control the expression level of mNeonGreen. These plasmids, which expressed mNeonGreen-tagged mutant FtsZ and mutant FtsZ at a ratio of ~1:1,

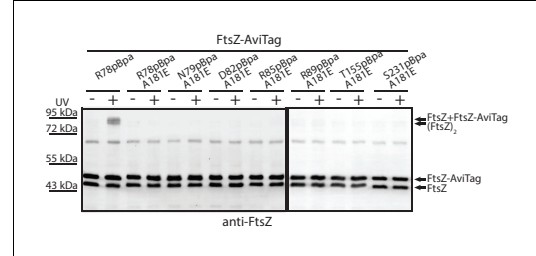

**Figure 6.** Protofilament formation is required for FtsZ lateral interactions to occur. Streptavidin blotting analyses of photocrosslinked products of EcFtsZ variants in which pBpa was incorporated at the lateral interface in addition to replacement of alanine 181 with glutamate, which is known to disrupt protofilament formation. The R78pBpa variant (with no disrupting replacement at position 181) was analyzed as a positive control.

DOI: https://doi.org/10.7554/eLife.35578.013

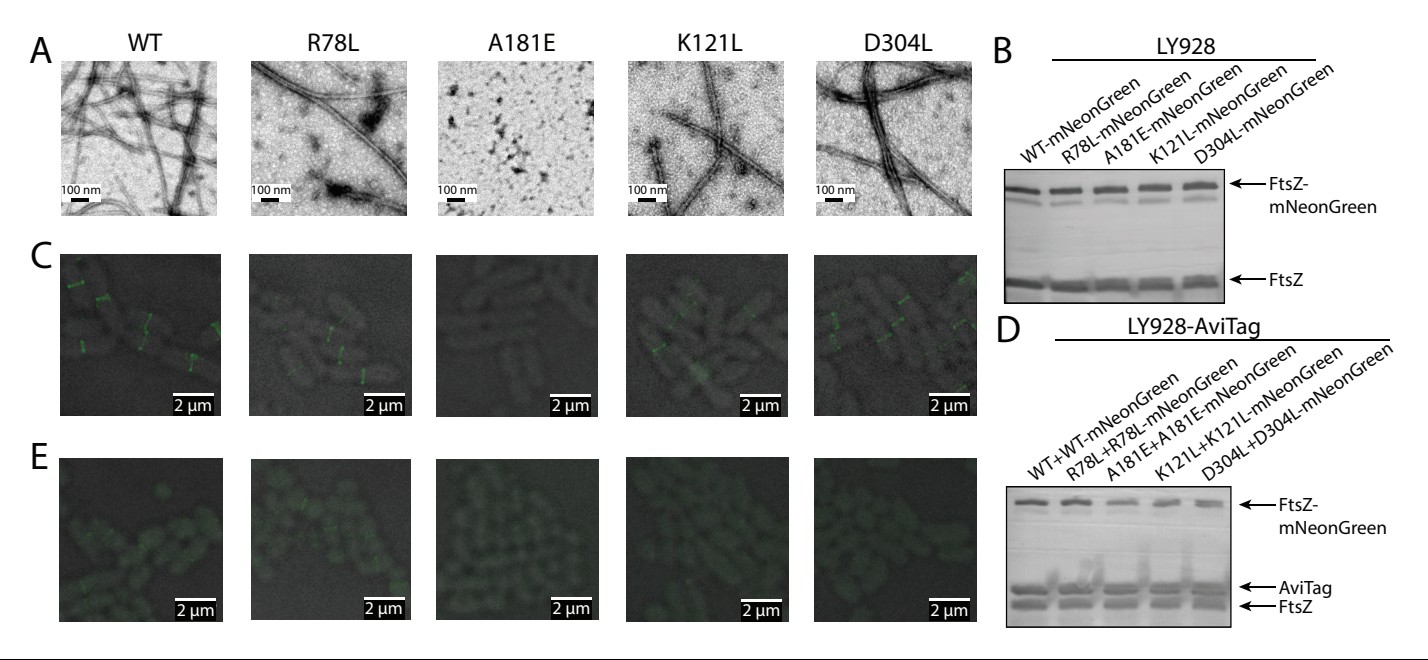

**Figure 7.** Disruption of the integration of lateral interaction-impaired protofilaments into the Z-ring. (**A**) Electron microscopy analysis of GTP-dependent polymerization (with the addition of 0.6 mg/ml DEAE-Dextran) of wild-type or mutated EcFtsZ in which all mutants except A181E form protofilaments similar to wild-type. (**B**) Western blot analysis of levels of mNeonGreen fusions to wild-type EcFtsZ and mutants. The cellular proportions of mutant FtsZ were ~40%. (**C**) Fluorescence microscopy demonstrates that low-level expression of laterally disruptive EcFtsZ mutants fused to mNeonGreen does not affect protofilament integration into the Z-ring. (**D**) Western blot analysis of levels of mNeonGreen fusions to wild-type EcFtsZ and mutants, as well as the AviTagged version. The cellular proportions of mutant FtsZ (mNeonGreen-tagged and untagged) were ~60%. (**E**) Dominant expression of disruptive EcFtsZ mutants fused to mNeonGreen hinders protofilament integration into the Z-ring, unlike wild-type or the non-disruptive interfacial mutant EcFtsZ[R78L].

DOI: https://doi.org/10.7554/eLife.35578.014

The following figure supplements are available for figure 7:

**Figure supplement 1.** Laterally disruptive FtsZ subunits incorporate into FtsZ protofilaments following a binomial distribution.
DOI: https://doi.org/10.7554/eLife.35578.015

**Figure supplement 2.** The relationships between the fraction of laterally disruptive subunits incorporated into the Z-ring, the cellular proportion of disruptive subunits, and the maximum number of disruptive subunits that still allow for efficient Z-ring lateral association.
DOI: https://doi.org/10.7554/eLife.35578.016

were transformed into *E. coli* LY928-*ftsz-avi* cells (*Figure 7D*, Materials and methods). The cellular proportions of mutant FtsZ (mNeonGreen tagged and untagged) increased to ~60% (as shown in *Figure 7D*), while the total levels of mNeonGreen-tagged FtsZ and untagged FtsZ (wild-type and mutant) were expressed at similar levels as before (*Figure 7B*). Fluorescence imaging demonstrated that midcell fluorescence was reduced to virtually undetectable levels for the laterally defective mutants, D304L and K121L(*Figure 7E*). By contrast, Z-rings remained visible in cells expressing a high level of the laterally non-disruptive mutant R78L (*Figure 7E*). Collectively, these results strongly suggest that lateral interactions are important for FtsZ protofilament assembly into the Z-ring.

## Discussion

FtsZ and its eukaryotic counterpart tubulin share a similar overall structure and use similar longitudinal interfaces to form protofilaments. However, unlike tubulin, which exhibits strong lateral interactions to form multi-stranded microtubules, FtsZ polymerizes into single-stranded protofilaments (*Mukherjee and Lutkenhaus, 1994*; *Romberg et al., 2001*), and undergoes GTP hydrolysis-driven treadmilling (*Bisson-Filho et al., 2017*; *Chen and Erickson, 2005*; *Loose and Mitchison, 2014*; *Mukherjee and Lutkenhaus, 1994*; *Yang et al., 2017*). These FtsZ protofilaments further coalesce and attach to the membrane at the division site through ZipA and FtsA (*Hale and de Boer, 1997*;

*Pichoff and Lutkenhaus, 2002*), forming the Z-ring. The role of lateral interfaces between FtsZ protofilaments in Z-ring dynamics is fundamental to our understanding of Z-ring function in bacterial cytokinesis. FtsZ protofilaments associate laterally to form higher-order polymers in vitro (*Bramhill and Thompson, 1994*; *Erickson et al., 1996*; *González et al., 2003*; *Löwe and Amos, 1999*; *Löwe and Amos, 2000*; *Oliva et al., 2003*; *Yu and Margolin, 1997*), and several studies have strongly suggested that lateral interfaces between protofilaments are important for FtsZ function (*Dajkovic et al., 2008*; *Milam et al., 2012*). In this study, we directly observed two different lateral interfaces of FtsZ protofilaments based on crystallographic analysis. We subsequently confirmed the presence of these lateral interfaces in living cells via in vivo photocrosslinking. Finally, we demonstrated that these weak, yet functionally important, lateral interfaces are involved in Z-ring assembly.

Lateral interfaces between FtsZ protofilaments have been extensively probed for decades and several residues have been genetically implicated in lateral interactions (*Haeusser et al., 2015*; *Jaiswal et al., 2010*; *Koppelman et al., 2004*; *Lu et al., 2001*; *Márquez et al., 2017*; *Moore et al., 2017*; *Shin et al., 2013*; *Stricker and Erickson, 2003*). However, some results from these studies were ambiguous and open to conflicting interpretations. Certain EcFtsZ mutants, such as E93R (*Jaiswal et al., 2010*), L169R (*Haeusser et al., 2015*), and D86K (*Lu et al., 2001*), have elevated tendency to form protofilament bundles in vitro, but evidence of their function in vivo remains inconclusive (*Stricker and Erickson, 2003*). Other EcFtsZ surface residues, such as G124 and R174, were identified as potential lateral residues since insertions of a fluorescent protein at these sites caused loss of function in vivo (*Koppelman et al., 2004*; *Moore et al., 2017*). However, insertion at R174 did not interfere with protofilament bundling in vitro, and insertion at G124 induced protein misfolding (*Moore et al., 2017*), making it challenging to link loss of function to defects in lateral interactions. Our structures and random photocrosslinking results also failed to provide cross-verification for any of these residues. Our observation that lateral interactions are weak in nature may offer an explanation for the ambiguities from these studies. The association of FtsZ protofilaments due to additive effect of lateral interactions between two protofilaments means that single mutant studies can suffer from insensitivity in vivo and in vitro. For example, our in vivo complementation studies revealed that, among the ten rationally designed, potentially disruptive mutants, eight of them were able to complement. In addition, in vitro protofilament bundling relies heavily on bundling agents such as $Ca^{2+}$ (*Löwe and Amos, 1999*), DEAE-dextran (*Erickson et al., 1996*), or Ficoll 70 (*González et al., 2003*). By contrast, our in vivo photocrosslinking study is able to unambiguously distinguish true lateral interactions from artifacts.

Another alternative explanation for these ambiguous results is other cellular factors that regulate interactions between FtsZ protofilaments in vivo. ZipA and various Zap proteins (ZapA, ZapC and ZapD) have been reported to crosslink, or to promote the lateral interactions between FtsZ protofilaments (*Durand-Heredia et al., 2011*; *Gueiros-Filho and Losick, 2002*; *Haeusser et al., 2015*; *Hale et al., 2000*; *2011*; *Huang et al., 2013*). Such enhancement of lateral association might enable ZipA/Zap to compensate for some intrinsic defects in lateral interactions between FtsZ protofilaments, thereby resulting in normal function for some of the mutants predicted to be disruptive. The residues involved in lateral association enhancement may be distinct from our findings. For example, the EcFtsZ L169R mutant can fully rescue the cell division defect of ΔzapAC cells (*Haeusser et al., 2015*), but L169 is far apart from the lateral interfaces we identified.

The inherently weak lateral interactions are unlikely to mediate formation of higher-order protofilament structures such as those observed in vitro. A recent electron cryotomography study showed a mean interprotofilament spacing of 6.8 nm, slightly too far apart to support tight interactions between FtsZ protofilaments (*Szwedziak et al., 2014*). Moreover, membrane-targeting FtsZ (either FtsZ and FtsA, or FtsZ alone with the addition of a membrane-targeting sequence) is sufficient to reconstitute contractile Z-rings in liposomes (*Osawa et al., 2008*; *Osawa and Erickson, 2013*; *Szwedziak et al., 2014*), and to display treadmilling behavior and the reorganization of FtsZ protofilaments into dynamic vortices on supported membranes (*Loose and Mitchison, 2014*; *Ramirez et al., 2016*). The antiparallel protofilament arrangement would preclude treadmilling of FtsZ. It is noteworthy that the cooperative assembly of single-stranded FtsZ protofilaments implies that the mechanism of FtsZ treadmilling is distinct from that of actin, for which lateral interactions are not directly involved. We propose that transient lateral interactions induce changes in treadmilling velocities of single protofilaments when they collide, rather than mediate the formation of a stable and static higher order architecture. Although the precise details of Z-ring dynamics remain to

be determined, this study is a vital step toward understanding the architecture and assembly mechanism of the bacterial cell division machinery in living cells, and provides novel structural information to guide the development of novel antimicrobial compounds that specifically target the division machinery.

## Materials and methods

### Key resources table

| Reagent type (species) or resource | Designation | Source or reference | Identifiers | Additional information |
|---|---|---|---|---|
| Gene (*Escherichia coli*) | FtsZ | NA | Uniprot ID: P0A9A6 | |
| Gene (*Mycobacterium tuberculosis*) | FtsZ | NA | Uniprot ID: P9WN95 | |
| Gene (*Branchiostoma lanceolatum*) | mNeonGreen | PMID:23524392 | Uniprot ID: A0A1S4NYF2 | |
| Stain, strain background (*Escherichia coli*) | W3110 *ftsZ*::kan recA56 Δ*ftsZ* | PMID:2045370 | | JKD7-1/pKD3, *ftsZ* conditional null strain with pKD3 rescue plasmid |
| Stain, strain background (*Escherichia coli*) | BW25113 Δ*insH11*:: pBpa-tRNA synthetase /tRNA-pBpa | PMID:27298319 | | |
| Stain, strain background (*Escherichia coli*) | BW25113 Δ*insH11*:: pBpa-tRNA synthetase /tRNA-pBpa Δ*ftsZ* | This paper | | *ftsZ* conditional null strain with pBpa incorporation system |
| Stain, strain background (*Escherichia coli*) | BW25113 Δ*insH11*::pBpa-tRNA synthetase/tRNA-pBpa *ftsZ*::*ftsZ-AviTag* | This paper | | The endogenous *ftsZ* gene was modified to encode FtsZ linked with an AviTag at the C-terminus |
| Antibody | Rabbit anti-EcFtsZ polyclonal antibody | This paper | | The antibody was harvested by immuning rabbit with the purified EcFtsZ protein. Dilution: 1:10000, for western blot |
| Antibody | Alkaline phosphatase-conjugated streptavidin | Beyotime | A0312 | Dilution: 1:5000, for western blot |
| Recombinant DNA reagent | pJSB100 | PMID:12896999 | | pJSB2-FtsZ |
| Recombinant DNA reagent | pTet-FtsZ | This paper | | *bla* pBR322 P tet::*ftsZ*, P tet incates Tet-on/Tet-off promoter and the expression of FtsZ in this plasmid via leaky expression |
| Recombinant DNA reagent | pTac-L3-FtsZ-mNeonGreen | This paper | | *bla* pBR322 P L3::*ftsZ-mNeonGreen*, P L3 incates a synthetic constitutive promoter selected from Anderson promoter collection |
| Recombinant DNA reagent | pTac-0.16-FtsZ-TAG-mNeonGreen | This paper | | *bla* pBR322 P0.16::*ftsZ-TAG-mNeonGreen*, P0.16 indicates a synthetic constitutive promoter selected from Anderson promoter collection and TAG amber condon was inserted into FtsZ-TAG-mNeonGreen fusion protein |
| Chemical compound, drug | *p*-benzoyl-L-phenylalanine (pBpa) | Bachem | F-2800 | |

### Protein expression, purification, and crystallization

The full-length *ftsZ* gene was amplified from *M. tuberculosis* genomic DNA and was subcloned into the pET15b plasmid vector. MtbFtsZ protein was overexpressed in BL21(DE3)/pLysS *E. coli* cells, cultured at 37°C in lysogeny broth (LB) medium and induced with 1 mM isopropyl β-D-1-thiogalactopyranoside (IPTG) after $OD_{600}$ reached 0.5. The His-tagged MtbFtsZ protein was then purified with Cobalt affinity resin. After removal of the His tag by thrombin cleavage, the protein was subjected

to size-exclusion chromatography performed with a Superdex 200 10/300 GL column (GE Health Sciences) that was pre-equilibrated with a buffer of 100 mM KCl, 0.1 mM EDTA, 20 mM Tris, pH 8.0, and 10% glycerol. The protein was concentrated to 20 mg/mL (as measured by ultraviolet absorbance), with 10 mM GTP added 30 min before crystallization. Well-diffracting crystals were grown by the sitting-drop vapor-diffusion method, in which 2 μL of the above MtbFtsZ-GTP solution were mixed with an equal volume of crystallization solution containing 1 M sodium citrate and 0.1 M imidizole, pH 8.0.

## Data collection and structure determination

Crystals were cryo-protected from their mother liquid by adding 30% glycerol, and were frozen in liquid nitrogen. Diffraction data were collected at the Shanghai Synchrotron Radiation Facility BL19U beamline (Shanghai, China). The data were indexed, integrated, and scaled using HKL-2000 (*Otwinowski and Minor, 1997*). Crystals are in space group $P6_522$ and contain three GDP-MtbFtsZ subunits per asymmetric unit. The best crystal diffracted X-rays to 2.7 Å resolution, with unit-cell dimensions of a = 100.5 Å, c = 138.3 Å. Phases were determined by molecular replacement using PHASER (*McCoy et al., 2005*) with the MtbFtsZ monomer (molecule A, PDB ID 1RQ7) (*Leung et al., 2004*) as a search model. Model adjustment was performed iteratively using Xtalview (*McRee, 1999*), and structure refinement was performed using REFMAC (*Collaborative Computational Project, Number 4, 1994*). The models were refined with data to 2.7 Å resolution, maintaining a highly restrained stereochemistry. The final model contains an FtsZ molecule and a GTP molecule. All structural illustrations were prepared with PYMOL (www.pymol.org).

## Characterization of EcFtsZ mutants via complementation assay

The complementation assay used here is based on the JKD7-1/pKD3 conditional null strain (*Dai and Lutkenhaus, 1991*) and the pJSB100 complementation vector (*Stricker and Erickson, 2003*). JKD7-1 is an *ftsZ*-null strain that is maintained in the presence of the rescue plasmid pKD3 that contains a functional *ftsZ* allele. The pKD3 plasmid is temperature sensitive for its replication, such that it is lost in a majority of the transformed *E. coli* cells when cultured at 42°C. The pJSB100 plasmid, derived from the pBAD vector, was used to express the wild-type or mutant EcFtsZ protein at a moderate level upon induction by arabinose. When strains containing both the pKD3 and the pJSB100 plasmids are grown at 42°C in the presence of arabinose, pKD3 fails to replicate and thus the survival of the cells relies on the expression of a functional EcFtsZ variant from pJSB100.

The complementation assay was performed as follows. JKD7-1/pKD3 cells were transformed with pJSB2 (carrying no *ftsZ* gene, as a negative control), pJSB100 (carrying the wild type *EcftsZ* gene, as a positive control), or a particular pJSB100-*EcftsZ* variant. The transformed cells were cultured in the repression medium (LB containing 34 μg/mL chloramphenicol, 100 μg/mL ampicillin, and 0.2% glucose) overnight at 30°C, reaching an $OD_{600}$ of 1.0–2.0. Ten microliters of 10,000-fold dilutions of the overnight cultures were then plated either on induction plates containing 0.05% arabinose or on repression plates containing 0.2% glucose. Repression plates were then cultured at 30°C, and the induction plates were cultured at 42°C, to determine the number of colony forming units (CFUs). CFU values on the induction plates were normalized to the CFU values from the repression plates. A mutant was considered to complement the *ftsZ* conditional-null strain when an induction plate produced at least 80% as many colonies as the repression plate. For each variant, the complementation assay was repeated three times.

Liquid complementation assays were performed by culturing cells transformed with pJSB100-derived plasmids that express mutant EcFtsZ proteins at 30°C to an $OD_{600}$ of 0.5 in repression medium. These cultures were then diluted 5,000,000-fold and cultured at 42°C for 24 hr in induction medium containing 0.05% arabinose. The successful complementation of *ftsZ* conditional-null cells by an EcFtsZ variant was defined by the ability for transformed cells to grow to an $OD_{600}$ >0.5; failure of complementation was defined as lack of growth (i.e., no measurable turbidity after overnight growth).

## In vivo photo-crosslinking analysis

The unnatural amino acid (pBpa) incorporation system is based on a plasmid expressing orthogonal pBpa-tRNA synthetase/tRNA$^{pBpa}$ pairs in *E. coli* (*Ryu and Schultz, 2006*). In generating a

complementation system to screen for functional pBpa variants of FtsZ, we constructed the LY928-Δ *ftsZ* (pJSB100) conditional null strain, in which the optimized genes encoding the pBpa-tRNA synthetase and tRNA$^{pBpa}$ (*Guo et al., 2009*) are integrated into the chromosome and a functional FtsZ protein is expressed from pJSB100 upon arabinose induction (*Stricker and Erickson, 2003*).

For a pBpa variant of FtsZ that successfully rescued the growth of LY928-Δ*ftsZ* (pJSB100) in repression medium (LB containing 50 μg/ml ampicillin and 0.2% glucose), the encoding plasmid was transformed into the LY928-*ftsZ-avitag* strain (whose endogenous *ftsZ* gene was modified to encode FtsZ linked with an AviTag at the C-terminus). The transformed cells were then grown at 37°C to mid-log phase in repression medium supplemented with 100 μM pBpa. One milliliter was then transferred to a 1.5 mL Eppendorf tube, irradiated at room temperature with UV light (365 nm) for 10 min using a Hoefer UVC 500 Crosslinker installed with 365 nm UV lamps (Amersham Biosciences) at a distance of 3 cm. Cells were subsequently harvested by centrifugation at 13,000 × *g* for 5 min, added into the loading buffer, and boiled. The cell lysate was then analyzed by tricine SDS-PAGE, and probed either by immunoblotting with FtsZ antibody or with streptavidin-alkaline phosphatase conjugate. Gel bands were scanned and processed using GIMP.

## Photo-crosslinking screening of FtsZ variants with randomly inserted pBpa

A library of expression plasmids in which the amber codon was randomly substituted for any triplet nucleotide in the *ftsZ* gene was constructed using *E. coli* Top10 cells, based on a method modified from an earlier study (*Daggett et al., 2009*). The plasmid library was used to transform LY928-Δ*ftsZ* (pJSB100) to select for variants that complement the *ftsZ* null phenotype. These complementing variants were subjected to in vivo photo-crosslinking analysis, and were sequenced to identify the site of the TAG codon replacement. The resulting library contains in-frame TAG mutations only in the N-terminal domain of FtsZ. Since the plasmid is leaky, and the in-frame TAG amber codon is read as a stop codon in *E. coli* Top10 cells, the generation of such a library would result in expression of truncated FtsZ proteins in cells. A likely explanation is that in-frame TAG mutation in the C-terminal domain would result in a truncated FtsZ with only the N-terminal domain, which is dominant negative.

## Fluorescence microscopy

Plasmids constitutively expressing mutant FtsZ fused to mNeonGreen (*Shaner et al., 2013*), with or without an amber codon inserted in between, were transformed into LY928 cells (in which optimized genes encoding the pBpa-tRNA synthetase and tRNA$^{pBpa}$ (*Guo et al., 2009*) were integrated into the chromosome), or LY928-*ftsZ-avitag* cells (whose endogenous *ftsZ* gene was modified to encode FtsZ linked with an AviTag by a GSG linker at the C-terminus). The transformed cells were cultured at 37°C in LB (containing 50 μg/mL ampicillin and 100 μM pBpa) to mid-log phase. Cells were then loaded onto a glass dish (NEST Biotechnology) and covered with a cover glass. Images were acquired on an N-SIM imaging system (Nikon) at 30°C with a 100X/NA1.49 oil-immersion objective (Nikon) and 488 nm laser beam. The reconstructed images were further processed with NIS-Elements AR 4.20.00 (Nikon) and GIMP. For experiments in *Figure 7C*, we used plasmids with synthetic constitutive promoter P$_{L3}$ (selected from the Anderson promoter collection: parts.igem.org/Promoters/Catalog/Anderson) to express FtsZ-mNeonGreen fusion protein. For experiments in *Figure 7E*, we used plasmids with the synthetic constitutive promoter P$_{0.16}$ to express TAG inserted FtsZ-TAG-mNeonGreen. We used a counter-selective recombining technique based on lambda-Red recombination system to tag the *ftsz* gene in LY928 cells (*Lee et al., 2009*). The expression levels of FtsZ were determined by Western-blot, and the proportions of mutant FtsZ were measured by analyzing the images with ImageJ gel analysis tool (https://imagej.en.softonic.com/).

## Electron microscopy

For *Figure 1A and B*, MtbFtsZ or EcFtsZ proteins (1 mg/mL) were first incubated in MEMK6.5 buffer (100 mM morpholine ethane sulfonic acid, pH 6.5 adjusted with KOH, 1 mM EGTA, 5 mM Mg acetate) with the addition of 0.6 mg/mL DEAE-Dextran, and in the presence of 2 mM GTP. For *Figure 7A*, wild-type and mutant EcFtsZ proteins (1 mg/mL) were first incubated in MEMK6.5 buffer in the presence of 2 mM GTP. The reaction mixture was then incubated on ice for 5–10 min, then at

37°C for 5–10 min, before a 5 μL aliquot was placed on a glow-discharged carbon-coated copper grid and negatively stained with 2% aqueous uranyl acetate. The air-dried grids were subsequently examined with a HITACHI HT7700 transmission electron microscope operated at 80 kV, or with a FEI Tecnai-F20 transmission electron microscope operated at 200 kV. Images of FtsZ protein assemblies were acquired on a Gatan ORIUS CCD camera at a nominal magnification of 40,000X, or with a Gatan Ultra4000 CCD camera at a nominal magnification of 50,000X.

### Simulations

We performed simulations based on the model of Z-ring formation described below, and calculated the percentage of laterally disruptive FtsZ subunits incorporated into the Z-ring. We assumed that in order to incorporate into the Z-ring through lateral bonds, a protofilament loses translational and rotational degrees of freedom and hence there is an entropic cost for immobilizing a protofilament. Since only wild-type FtsZ subunits contribute to lateral attachment, the balance between the energy of binding and the entropic cost results in an upper limit to the fraction of laterally disruptive subunits that a protofilament can tolerate and still incorporate into Z-ring. To simplify, we consider the dynamics of 200 protofilaments in the simulations, and all protofilaments are set to be 50 subunits long. We then use a variable threshold $T$, which is related to the critical fraction ($f_c$) by: $f_c = T/50$. In each simulation, we set the overall proportion f of laterally disruptive subunits in a cell and generated a vector $\mathbf{X} = (x_1, x_2, ..., x_{200})$, where $x_i$ represents the number of laterally disruptive subunits in the $i^{th}$ protofilament, and was selected based on a Binomial distribution with probability f (*Figure 7—figure supplement 1*). For a protofilament with more or less laterally disruptive subunits than the threshold T, we set the probability of Z-ring incorporation to 0.01 or 0.99, respectively. We used a Boolean vector $\mathbf{V} = (0, 1, ..., 1)$ to represent the states of protofilaments, where 1 or 0 indicate incorporation or not into the Z-ring, respectively. We then calculated the percentage of laterally disruptive subunits incorporated into the Z-ring as:

$$\frac{\mathbf{X} \cdot \mathbf{V}}{\sum_{i=1}^{200} x_i}$$

For each value of f and T, we performed 10,000 independent simulations.

### Acknowledgements

We thank Harold Erickson for insightful comments and suggestions and the pJSB100 complementation vector, and Joe Lutkenhaus for discussions and critical reviews of the manuscript and the JKD7-1/pKD3 conditional null strain. We thank the Core Facilities at School of Life Sciences, Peking University for assistance with SIM, and Chunyan Shan and Xiaochen Li for assistance with fluorescence imaging. This work was supported in part by funds from the Ministry of Science and Technology (Awards 2016YFA0500404 and 2014CB910300 to SY and 2012CB917300 to ZYC), the National Natural Science Foundation of China (Awards 31525001 and 31430019 to SY, and 31670775 and 31470766 to ZYC), and the Fundamental Research Funds for the Central Universities (to SY). KCH is a Chan Zuckerberg Biohub Investigator. Structure coordinates and reflection files have been deposited in the protein data bank under accession number 5ZUE.

## Additional information

### Funding

| Funder | Grant reference number | Author |
|---|---|---|
| Ministry of Science and Technology of the People's Republic of China | 2016YFA0500404 | Sheng Ye |
| Ministry of Science and Technology of the People's Republic of China | 2014CB910300 | Sheng Ye |

| | | |
|---|---|---|
| Ministry of Science and Technology of the People's Republic of China | 2012CB917300 | Zengyi Chang |
| National Natural Science Foundation of China | 31525001 | Sheng Ye |
| National Natural Science Foundation of China | 31430019 | Sheng Ye |
| National Natural Science Foundation of China | 31670775 | Zengyi Chang |
| National Natural Science Foundation of China | 31470766 | Zengyi Chang |
| Fundamental Research Funds for the Central Universities | | Sheng Ye |

The funders had no role in study design, data collection and interpretation, or the decision to submit the work for publication.

### Author contributions

Fenghui Guan, Data curation, Methodology, Writing—original draft; Jiayu Yu, Data curation, Formal analysis, Methodology, Writing—original draft; Jie Yu, Yang Liu, Ying Li, Data curation; Xin-Hua Feng, Formal analysis; Kerwyn Casey Huang, Formal analysis, Writing—review and editing; Zengyi Chang, Data curation, Supervision, Funding acquisition, Methodology, Writing—original draft; Sheng Ye, Conceptualization, Supervision, Funding acquisition, Methodology, Writing—original draft, Writing—review and editing

### Author ORCIDs

Fenghui Guan (iD) http://orcid.org/0000-0002-2756-5736
Sheng Ye (iD) http://orcid.org/0000-0001-9300-6257

### Decision letter and Author response

Decision letter https://doi.org/10.7554/eLife.35578.021
Author response https://doi.org/10.7554/eLife.35578.022

## Additional files

### Supplementary files

• Transparent reporting form
DOI: https://doi.org/10.7554/eLife.35578.017

### Data availability

Diffraction data have been deposited in PDB under the accession code 5ZUE.

The following dataset was generated:

| Author(s) | Year | Dataset title | Dataset URL | Database, license, and accessibility information |
|---|---|---|---|---|
| Guan F, Yu J, Yu J, Liu Y, Li Y, Feng XH, Huang KC, Chang Z, Ye S | 2018 | GTP-bound, double-stranded, curved FtsZ protofilament structure | http://www.rcsb.org/pdb/search/structid-Search.do?structureId=5ZUE | Publicly available at the RCSB Protein Data Bank (accession no: 5ZUE) |

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
