## [Decision Letter]

[Editors’ note: a previous version of this study was rejected after peer review, but the authors submitted for reconsideration. The first decision letter after peer review is shown below.]

Thank you for submitting your work entitled "Lateral interactions between protofilaments of the bacterial tubulin homolog FtsZ that are essential for cell division" for consideration by *eLife*. Your article has been reviewed by three peer reviewers, and the evaluation has been overseen by a Reviewing Editor and a Senior Editor. The following individuals involved in review of your submission have agreed to reveal their identity: Joe Lutkenhaus (Reviewer #2).

Our decision has been reached after consultation between the reviewers. Based on these discussions and the individual reviews below, we regret to inform you that your work will not be considered further for publication in *eLife*.

While all three reviewers had many positive comments about the paper, each raised serious concerns that would preclude acceptance of this paper. It was felt that the revisions and additional experiments needed to address these concerns would be so major that they would make the submission of a revised paper within a short period of time impossible.

Reviewer #1:

This paper addresses a controversial issue about FtsZ assembly – lateral interactions. The conclusion is that lateral contacts occur and are functionally relevant. The crosslinking is well done however, I question that results in Figure 6, which are the most important since they provide evidence of physiological function. My concern comes down to 3 things. (1) I do not understand the lack of incorporation of the mutants described into the Z ring in vivo. In the experiments, the mutant represents a small fraction of the total FtsZ and should copolymerize so it should be incorporated into the Z ring by the preponderance of Wild type FtsZ. Perhaps, the combination of D304L or K121L with GFP causes some unexpected problem. (2) it appears that the mutants polymerize okay and even bundle in vitro (this is not mentioned in the text) based on the width of the polymers observed in the Figure. This suggests that the in vitro assembly is not a relevant assay or that the mutations do not affect bundling. However, it argues that the mutants should copolymerize with the WT FtsZ. (3) If the lateral contacts identified in this study are so important then one would expect the reciprocal mutants to also be defective. So, if D304L is lethal then the R229L or equivalent should also be lethal.

Reviewer #2:

This is an interesting manuscript that probes a fundamental question in bacterial cytokinesis – the importance of lateral interactions between FtsZ protofilaments and their contribution to the formation of a functional Z-ring in the cell. Most of the experiments are well designed and the article is written with clarity. Nonetheless, I have the following comments about the manuscript:

1) The authors show that R229 and D301 form salt-bridges in at least 2 of the 3-subunits in the double-stranded MtbFtsZ protofilament structure (Figure 1E). Yet, the leucine substitution at S231(R229) but not D304(D301) can yield cross-linking dimers (Figure 2) and complement in vivo (Figure 4 and 4S1). The authors need to further clarify the "essentiality" of D304(D301) in mediating lateral interactions. In the context of multiple weak electrostatic interactions contributing to lateral interactions between protofilaments as a whole, why are some residues essential and not others?

2) Based on the structural data, the authors conclude that residues R119(K121) and D301(D304) are critical in mediating lateral interactions. The inability of substitutions at K121 and D304 to form cross-linking dimers (Figure 5), complement in vivo (Figure 4), and the mutant protein fusions to be recruited to the Z-ring in vivo (Figure 6) are reported to be consistent with the importance of these residues in mediating lateral interactions in the cell. Additional evidence beyond the crystallographic interface data that K121 and D304 are indeed mediating lateral interactions is warranted to make a more convincing argument. Also, is the implication that the R119 and R76 side-chains hydrogen bond in Figure 1D inset?

3) Why do lateral interaction mutants that form protofilaments (K121L or D304L) fail to integrate into the Z-ring? Presumably, in vivo, these protofilaments can still be tethered to the membrane by FtsA and/or ZipA, which interact with the C-terminal tail of FtsZ. It is conceivable that these mutants can associate via their second "functional" lateral interface with native FtsZ protofilaments and incorporate into a mixed Z-ring. Another possibility could be that these mutant protofilaments are crosslinked to native polymers through bundling proteins to generate mixed Z-rings in the cells.

4) The electron microscopy analyses of K121L and D304L mutants under GTP-dependent polymerization conditions appear to show similarly associated protofilaments as WT or R78L (a non-disruptive lateral interaction mutant) (Figure 6) – maybe the mutant protofilaments can associate using the second "functional" lateral interface under these conditions. Perhaps the authors could include some quantitative analysis of the EM images in terms of the thickness of the double-stranded protofilaments in the various mutants compared to the WT.

5) While the complementation assays as reported are reasonable, visualization of cell and Z-ring morphologies in the various mutants could provide meaningful differences between the various residues mediating lateral interactions.

6) The authors discuss diffusion dynamics of the various configurations of the protofilaments in their model (Figure 7), however, there is no discussion of how lateral interactions, especially in configuration 3 stated as "with stable lateral interactions holding neighboring protofilaments firmly in place", reconcile with the treadmilling of FtsZ protofilaments and the rapid turnover of individual FtsZ subunits in the cell.

Reviewer #3:

This paper addresses the role of higher-order filament architecture in the function of FtsZ, the tubulin-like cell division protein of bacteria that assembles into a cytokinetic ring. This is a question fundamental to our understanding of how the ring works to organize and drive cytokinesis. Previous work by Li et al., (2003) crystallized a double stranded antiparallel FtsZ filament of *Mycobacterium tuberculosis* (Mtb) and characterized residues in FtsZ important for longitudinal interactions within protofilaments (pfs). In the present study, some of the same authors have crystallized another antiparallel double stranded FtsZ filament from Mtb that has a different and more extensive inter-filament interface consisting mainly of charged residues. They cleverly use photocrosslinking with unnatural amino acids inserted at these and other residues to identify those involved in close interactions with another residue of FtsZ. This assay identifies critical charged residues most likely involved in lateral interactions between pfs. The authors then show that altering these residues (e.g. to leucine) in a few cases prevents the interactions and blocks in vivo function, including the ability to incorporate into the FtsZ ring.

While the study is important, generally well done and mostly clearly written, I do have a number of concerns that need to be addressed.

Major comments:

1) In this work, the authors have isolated another antiparallel double stranded FtsZ filament from Mtb that has a different inter-filament interface from the previously published structure by Li et al., 2003. Given that they rely heavily on the new pf structure for most of their mutant choices and for speculations about electrostatic interactions, they need to be clearer about why the new structure is different from the previous one, and why they think this new structure is more physiologically relevant.

2) It is puzzling that none of the residues found here at the lateral interfaces corresponds with residues genetically implicated in lateral interactions in previous reports: D86K (shown by Stricker and Erickson, 2003 to form paired pfs); R174D (originally found by Koppelman et al., 2004, to be defective in pf bundling, albeit recently disputed by Moore et al., 2016; E93R (shown by Jaiswal et al., 2010, to hyperbundle in vitro and fail to function in vivo) and L169R (shown by Haeusser et al., 2015, to hyperbundle in vitro and to bypass ZipA function in vivo). The authors need to mention this, hopefully with some kind of explanation. This would be a better use of Discussion section space (see comments about the Discussion section below).

3) It was surprising to see no mention whatsoever of ZipA and Zap proteins that are known, in some cases quite clearly, to promote lateral interactions between pfs (or at least crosslinking), and their potential roles (see also below). Furthermore, I don't think the model in Figure 8 is all that helpful in part because it does not consider the roles of these proteins in higher order assembly of FtsZ. Perhaps some of the leucine substitution mutants functioned normally for cell division is because Zap/ZipA can compensate for an intrinsic defect in lateral interactions between pfs.

4) It was hard to follow the different mutants, despite having the alignment figure. Could all the mutants (*E. coli* residue numbers) be shown on the crystal structure of the pfs? Along the same lines, it should be made more clear up front that the crosslinking studies were done with *E. coli* FtsZ.

The Discussion section was disappointing for a number of reasons outlined below:

1) Although concise and well written, it is superficial and somewhat speculative and refers to a vague cartoon model that does not provide much new insight into how FtsZ protofilaments might work in the cell. The potential contributions of interfaces 1 and 2 are not discussed either. Finally, the recent evidence that FtsZ pfs move by treadmilling (Yang et al., and Bisson Filho et al., 2016) and how that activity relates to pf lateral interactions was not mentioned in the text or incorporated into the model.

2) Electrostatic pairing is proposed as the main mechanism for lateral interactions between protofilaments, based on the crystal structure and the predominance of charged residues at the interface. This may be true, but there is very little additional supporting evidence for this claim other than a few substitutions with leucine are disruptive. It does not help the case that one of the critical residues at interface 1 of *E. coli* FtsZ is a serine, not an arginine as in Mtb FtsZ. For stronger proof that electrostatics are involved, the authors should at least show that charge swaps between two known interacting residues maintains lateral interactions and in vivo function.

3) Can the authors rule out the possibility that the formation of some crosslinked dimers is due to interactions with another (longitudinal) subunit within the pf instead of a lateral interaction with an adjacent pf? Insertion of pBpa at the longitudinal interface residue K140 seems to allow crosslinking (Figure 3B), but is that because pBpa at residue 140 still allows function?

4)I found it surprising that 8 of the leucine replacements were able to complement. Perhaps single replacements have smaller effects on lateral interactions because other interactions in the interface compensate, but the explanation offered does not seem sufficient. Perhaps changes to the opposite charge would have a larger negative effect?

5) There is no mention anywhere in the manuscript (or model in Figure 8) of the role of ZipA or Zap proteins in promoting lateral interactions between FtsZ protofilaments. The nice in vivo crosslinking results may result in part from the action of these and other FtsZ-bundling proteins and not solely from intrinsic ability of FtsZ protofilaments to interact laterally. This should be explored in the Discussion section.

6) It is great that the crosslinking strategy and data were internally consistent and were consistent with the functional results. However, given that ZipA and Zap proteins (among others) may influence lateral interactions between FtsZ protofilaments in vivo, the argument that the interfaces found here are important for intrinsic lateral interactions would be strengthened by testing the ability of the purified FtsZ mutants to bundle in vitro. I suggest choosing a couple of FtsZ mutants that fail to crosslink in vivo and subjecting the purified proteins to the in vitro bundling conditions used in Figure 1A-B; the prediction is that they would fail to form bundles. They already have purified K121L and D304L that were shown in Figure 6B to form protofilaments under non-bundling conditions (i.e. no DEAE-dextran), so this should be an easy experiment

7) Nevertheless, from the EM images in Figure 6B, it looks like K121L and D304L do form some paired filaments, which they shouldn't at all if they are defective in lateral interactions. Can the authors explain this contradiction?

[Editors’ note: what now follows is the decision letter after the authors submitted for further consideration.]

Thank you for resubmitting your article "Lateral interactions between protofilaments of the bacterial tubulin homolog FtsZ are essential for cell division" for consideration by *eLife*. Your article has been reviewed by two peer reviewers, and the evaluation has been overseen by a Reviewing Editor and Michael Marletta as the Senior Editor. The following individuals involved in review of your submission have agreed to reveal their identity: Joe Lutkenhaus (Reviewer #2).

The reviewers have discussed the reviews with one another and the Reviewing Editor has drafted this decision to help you prepare a revised submission. Both reviewers agreed that the paper has improved since the original decision, but both reviewers felt also that the paper cannot yet be accepted in its current form.

Summary:

This resubmission is an improvement over the original. It puts previous work about FtsZ bundling in better context, including other FtsZ-bundling factors such as Zap proteins, and clarifies some of the other specific concerns voiced by the first round of reviewers. Although previous microscopic and genetic evidence for lateral interactions between FtsZ filaments is quite strong, the main novelty of this study is that it is the first biochemical confirmation of lateral interactions at the atomic level between FtsZ protofilaments in cells. It also suggests that residues important for lateral interactions in vivo are different from those implicated in formation of sheets, ribbons, and bundles in vitro. This discrepancy can be rationalized by the presence of Zap proteins in vivo that keep FtsZ filaments close together, and by other proteins, e.g. FtsA, that were recently shown to do the opposite (i.e. keep FtsZ filaments apart) in a reconstituted system (PMID 28695917).

Essential revisions:

1) Since that first submission over a year ago, the two Science papers on FtsZ treadmilling in *E. coli* and *B. subtilis* were published, establishing the importance of FtsZ filament directionality at the Z-ring. In addition, Yao et al., (2017) reported additional cryo-electron tomographic evidence for close associations between FtsZ filaments in cells. While Yao et al., argue further that FtsZ lateral interactions occur in vivo, that is merely one more piece of evidence for this. Rather, it is treadmilling of FtsZ filaments in *E. coli* cells that provides the biggest concern: How can antiparallel double filaments, which the authors here, claim are forming in cells, treadmill? Treadmilling requires filaments, whether single or in a group, to have the same directionality. Antiparallel double filaments of MreB, for instance, are not expected to treadmill, and they don't. How can the authors reconcile FtsZ's important treadmilling activity with their antiparallel filament model?

In light of this, an alternative explanation for the authors' data is that there are many close interactions between FtsZ protofilaments that occur in vivo, potentially many which are transient in nature, and they have found a subset of them.

2) A number of concerns were raised about Figure 7A, which shows in vitro assembly of FtsZ filaments by EM. The first is that there is no scale bar, which makes it hard to evaluate whether the filaments are single or double. Second, the field of view is so small that it is hard to make any conclusions about the overall FtsZ assembly state. But most importantly, it is clear that the D304L is forming some type of bundles or at least double stranded filaments. The authors responded to this concern in their rebuttal by discussing that "addition of the bundling agents is likely to be critical for the observed effects". However, no bundling agents were added for this experiment, and other FtsZ mutants with enhanced lateral interactions clearly have been reported to form at least double stranded filaments with no added bundling agents (see the Discussion section, which seems to ignore these results). This is in contrast to mostly single stranded filaments formed by wild type FtsZ under these conditions. If these mutant FtsZs were truly defective in lateral interactions, they should form only single stranded filaments, even perhaps in the presence of bundling agents. It is difficult to reconcile the seemingly enhanced lateral interactions observed with D304L in EM with the defective lateral interactions in vivo; the authors need to do a better job of presenting the data and discussing it.

3) The failure of the D304L and K121L mutant FtsZs to incorporate into the Z-ring described in Figure 7 is backed by some reasonably convincing theoretical data, but still is not explained well enough for general readers. The Materials and methods section does not adequately describe the procedure. In subsection “Fluorescence microscopy”, there is no indication of any inducer added, but yet in subsection “Lateral interfaces are involved in Z-ring assembly” it is mentioned that an inducer was used. How do the levels of the mutant FtsZs increase? The legend to Figure 7 needs to provide more detail about the three proteins being expressed and how they are expressed, instead of making conclusions from the data.

4) The critical experiments are in Figure 7 where the authors test assembly of 3 of the FtsZ mutants in vitro and incorporation into the Z ring in vivo (2 mutants affected for crosslinking and one that is not). Panel A shows the assembly in vitro. Part of this is to show the mutant can assemble. However, the morphology of the polymers show that they appear to bundle as well as the wild type (looks like this from the dimensions of the polymers). This is of some concern but as said above the correlation between in vitro bundling and what goes on in vivo is uncertain. In panel C the authors show that the mutant gets incorporated in the Z ring when ectopically expressed at a low level. In panel E the authors do a rather complicated experiment. The idea is to see if the mutant protein gets incorporated into the ring if expressed at a higher level. The results are not so convincing, in part because this is a negative result (no incorporation) rather than a positive. The idea is that if the mutant protein copolymerizes with the WT protein there will be a mixture of filaments with variable amount of WT and mutant subunits. The argument is that if a filament has a sufficient number of mutant subunits it may be defective for lateral interaction and not be incorporated. However, there still has to be enough filaments with sufficient proportion of WT subunits so that a Z ring is formed. I find it hard to believe that adding a mutant to a level that it would not get incorporated would not cause some trouble for cell division. Division is sensitive to the level of FtsZ (30% drop blocks division) and some of the WT FtsZ has to be trapped in filaments that are nonfunctional (can't get incorporated into the Z ring) it should cause trouble. The addition of the mutant FtsZ will titrate some of the WT FtsZ away interfering with cell division. From the images shown there does not appear to not be problem as the cells appear shorter than in panel C.

Two things were suggested to make this more convincing. In subsection “Lateral interfaces are involved in Z-ring assembly” (and in Figure 7E) the authors talk about a pool of intermediate size (referring to the laterally disruptive FtsZs). It follows from this that a pool of larger size should be dominant negative. This would be a simple and easy test. If the mutants behave as the authors think then overexpression of the mutants in wild type cells should destroy the Z ring whereas the control mutant should not. This should be easy to see. Also, induction of the nonaffected mutant as a control is important since overexpression of FtsZ will cause filamentation at some point. Nevertheless, the lateral mutants should be much more toxic. The authors could just see the cells filament when the mutant protein is induced. They could use a strain with ZapA-GFP and see the ring disappear when the mutant is induced. A second experiment that should be done is to express the mutants in a strain where WT ftsZ is depleted and see what structures the mutants make. Do they form a ring on their own? This could be assessed by immunofluorescence microscopy.

---

## [Author Response]

[Editors’ note: the author responses to the first round of peer review follow.]

Reviewer #1:This paper addresses a controversial issue about FtsZ assembly – lateral interactions. The conclusion is that lateral contacts occur and are functionally relevant. The crosslinking is well done however, I question that results in Figure 6, which are the most important since they provide evidence of physiological function. My concern comes down to 3 things. (1) I do not understand the lack of incorporation of the mutants described into the Z ring in vivo. In the experiments, the mutant represents a small fraction of the total FtsZ and should copolymerize so it should be incorporated into the Z ring by the preponderance of Wild type FtsZ. Perhaps, the combination of D304L or K121L with GFP causes some unexpected problem. (2) it appears that the mutants polymerize okay and even bundle in vitro (this is not mentioned in the text) based on the width of the polymers observed in the Figure. This suggests that the in vitro assembly is not a relevant assay or that the mutations do not affect bundling. However, it argues that the mutants should copolymerize with the WT FtsZ. (3) If the lateral contacts identified in this study are so important then one would expect the reciprocal mutants to also be defective. So, if D304L is lethal then the R229L or equivalent should also be lethal.

We appreciate the reviewer’s positive evaluation of our work and its significance.

Below, we address the three major concerns raised by the reviewer point by point.

1) We performed new fluorescence microscopy experiments by carefully controlling the expression level of the mutant proteins. We observed that when the fraction of laterally disruptive FtsZs is low, most protofilaments can tolerate the disruption and incorporate into the Z-ring. By contrast, when the fraction is high, the laterally disruptive FtsZs become dominant negative and interfere with Z-ring formation. When the fraction is intermediate, the protofilaments that happen to have less laterally disruptive subunits are able to incorporate into the Z-ring, while others with more disruptive subunits cannot. As a result, raising the proportion of laterally disruptive FtsZs in the cell can dramatically reduce on the fraction of laterally disruptive subunits incorporated into the Z-ring, resulting in a faint or undetectable fluorescence signal from the labeled disruptive subunits, as we show in the new Figure 7. These points are further highlighted by our simulations described in the supplements to Figure 7.

2) We confirmed that the FtsZ mutant proteins D304L and K121L are still capable of forming GTP-dependent protofilaments (Figure 7A). These data indicate that when these laterally disruptive FtsZ protein are co-expressed with wild-type FtsZ, they can stochastically copolymerize to form mixed incorporated protofilaments.

3) We agree with the reviewer that the complementation results of the disruptive mutants at Ser231 and Asp304, two residues likely to be involved in direct interactions on lateral interface 1, were dramatically different. This discrepancy raised an obvious concern as to whether Asp304 is important for other functions. To address this concern, we performed additional complementation studies by generating double mutants based on D304L and observed that two double mutants across the interface (D304L/S231E and D304L/S231Q) resulted in complementation, strongly indicating that lateral interface 1 does form in vivo (Figure 5).

Reviewer #2:This is an interesting manuscript that probes a fundamental question in bacterial cytokinesis – the importance of lateral interactions between FtsZ protofilaments and their contribution to the formation of a functional Z-ring in the cell. Most of the experiments are well designed and the article is written with clarity. Nonetheless, I have the following comments about the manuscript:

We appreciate the reviewer’s positive evaluation of our work and its significance. Below, we address the concerns raised by the reviewer point by point.

1) The authors show that R229 and D301 form salt-bridges in at least 2 of the 3-subunits in the double-stranded MtbFtsZ protofilament structure (Figure 1E). Yet, the leucine substitution at S231(R229) but not D304(D301) can yield cross-linking dimers (Figure 2) and complement in vivo (Figure 4 and 4S1). The authors need to further clarify the "essentiality" of D304(D301) in mediating lateral interactions. In the context of multiple weak electrostatic interactions contributing to lateral interactions between protofilaments as a whole, why are some residues essential and not others?

We agree with the reviewer that the complementation results of the disruptive mutants at Ser231 and Asp304, two residues likely involved in direct interactions in lateral interface 1, were dramatically different. This discrepancy raised an obvious concern as to whether Asp304 is important for other functions. To address this concern, we performed additional complementation studies by generating double mutants based on D304L and observed that two double mutants across the interface (D304L/S231E and D304L/S231Q) resulted in complementation, strongly indicating that lateral interface 1 does form in vivo (Figure 5).

2) Based on the structural data, the authors conclude that residues R119(K121) and D301(D304) are critical in mediating lateral interactions. The inability of substitutions at K121 and D304 to form cross-linking dimers (Figure 5), complement in vivo (Figure 4), and the mutant protein fusions to be recruited to the Z-ring in vivo (Figure 6) are reported to be consistent with the importance of these residues in mediating lateral interactions in the cell. Additional evidence beyond the crystallographic interface data that K121 and D304 are indeed mediating lateral interactions is warranted to make a more convincing argument. Also, is the implication that the R119 and R76 side-chains hydrogen bond in Figure 1D inset?

We would like to clarify that our conclusion that residues R119(K121) and D301(D304) are critical in mediating lateral interactions is not based simply on our structural data. The R119 and R76 side chains hydrogen bond in the Figure 1D inset only suggest that they might be important. Structural data allowed us to identify 10 interfacial residues that we tested directly in cross-linking and complementation studies. These studies allowed us to conclude that the two residues are critical in mediating lateral interactions.

Our additional complementation studies with double mutants based on D304L discussed above strongly support our conclusion that lateral interface 1 does form in vivo (Figure 5).

3) Why do lateral interaction mutants that form protofilaments (K121L or D304L) fail to integrate into the Z-ring? Presumably, in vivo, these protofilaments can still be tethered to the membrane by FtsA and/or ZipA, which interact with the C-terminal tail of FtsZ. It is conceivable that these mutants can associate via their second "functional" lateral interface with native FtsZ protofilaments and incorporate into a mixed Z-ring. Another possibility could be that these mutant protofilaments are crosslinked to native polymers through bundling proteins to generate mixed Z-rings in the cells.

We performed new fluorescence microscopy experiments by carefully controlling the expression level of the mutant proteins. We observed that when the fraction of laterally disruptive FtsZs is low, most protofilaments can tolerate the disruption and incorporate into the Z-ring. By contrast, when the fraction is high, the laterally disruptive FtsZs become dominant negative and interfere with Z-ring formation. When the fraction is intermediate, the subset of protofilaments with less laterally disruptive subunits are able to incorporate into the Z-ring, while others with more disruptive subunits cannot. As a result, raising the proportion of laterally disruptive FtsZs in the cell can dramatically reduce the fraction of laterally disruptive subunits incorporated into the Z-ring, resulting in a faint or undetectable fluorescence signal from the labeled disruptive subunits, as we show in the new Figure 7. These points are further highlighted by our modeling studies in the supplements to Figure 7.

4) The electron microscopy analyses of K121L and D304L mutants under GTP-dependent polymerization conditions appear to show similarly associated protofilaments as WT or R78L (a non-disruptive lateral interaction mutant) (Figure 6) – maybe the mutant protofilaments can associate using the second "functional" lateral interface under these conditions. Perhaps the authors could include some quantitative analysis of the EM images in terms of the thickness of the double-stranded protofilaments in the various mutants compared to the WT.

in vitroassembly of FtsZ to form higher-order polymers requires nonphysiological conditions, such as Ca^2+^ (Lowe and Amos, 2017), DEAE dextran (Erickson et al., 1996), or Ficoll 70 (Gonzalez et al., 2003). Therefore, the addition of the bundling agents is likely to be critical for these effects. By contrast, our in vivo photocrosslinking studies are able to unambiguously distinguish true lateral interactions from artifacts. We have included these points in the Discussion section.

5) While the complementation assays as reported are reasonable, visualization of cell and Z-ring morphologies in the various mutants could provide meaningful differences between the various residues mediating lateral interactions.

We agree and are currently attempting to perform these studies. While such studies are well beyond the scope of our current manuscript, we look forward to the insights they may provide.

6) The authors discuss diffusion dynamics of the various configurations of the protofilaments in their model (Figure 7), however, there is no discussion of how lateral interactions, especially in configuration 3 stated as "with stable lateral interactions holding neighboring protofilaments firmly in place", reconcile with the treadmilling of FtsZ protofilaments and the rapid turnover of individual FtsZ subunits in the cell.

We have significantly modified the Discussion section and our model in accordance with his/her points.

Reviewer #3:This paper addresses the role of higher-order filament architecture in the function of FtsZ, the tubulin-like cell division protein of bacteria that assembles into a cytokinetic ring. This is a question fundamental to our understanding of how the ring works to organize and drive cytokinesis. Previous work by Li et al., (2003) crystallized a double stranded antiparallel FtsZ filament of Mycobacterium tuberculosis (Mtb) and characterized residues in FtsZ important for longitudinal interactions within protofilaments (pfs). In the present study, some of the same authors have crystallized another antiparallel double stranded FtsZ filament from Mtb that has a different and more extensive inter-filament interface consisting mainly of charged residues. They cleverly use photocrosslinking with unnatural amino acids inserted at these and other residues to identify those involved in close interactions with another residue of FtsZ. This assay identifies critical charged residues most likely involved in lateral interactions between pfs. The authors then show that altering these residues (e.g. to leucine) in a few cases prevents the interactions and blocks in vivo function, including the ability to incorporate into the FtsZ ring.While the study is important, generally well done and mostly clearly written, I do have a number of concerns that need to be addressed.

We appreciate the reviewer’s positive evaluation of our work and its significance. Below, we address the concerns raised by the reviewer point by point.

Major comments:1) In this work, the authors have isolated another antiparallel double stranded FtsZ filament from Mtb that has a different inter-filament interface from the previously published structure by Li et al., 2003. Given that they rely heavily on the new pf structure for most of their mutant choices and for speculations about electrostatic interactions, they need to be clearer about why the new structure is different from the previous one, and why they think this new structure is more physiologically relevant.

We would like to clarify that we are not proposing that the new structure is more physiologically relevant. Instead, each structure reveals an inter-protofilament interface. We observed an inter-protofilament interface from our previously published structure in Science. However, the existence of only a single lateral interface within such an antiparallel arrangement of protofilaments would be self-limiting and lead only to the formation of double-stranded filaments. Formation of bundles composed of more than two FtsZ protofilaments requires additional lateral interfaces between the opposite sides of the protofilaments. We have now identified a new inter-protofilament interface in a new hexagonal crystal of MtbFtsZ. Our studies show that both interfaces exist in living cells and are physiologically relevant.

2) It is puzzling that none of the residues found here at the lateral interfaces corresponds with residues genetically implicated in lateral interactions in previous reports: D86K (shown by Stricker and Erickson, 2003 to form paired pfs); R174D (originally found by Koppelman et al., 2004, to be defective in pf bundling, albeit recently disputed by Moore et al., 2016; E93R (shown by Jaiswal et al., 2010,, to hyperbundle in vitro and fail to function in vivo) and L169R (shown by Haeusser et al., 2015, to hyperbundle in vitro and to bypass ZipA function in vivo). The authors need to mention this, hopefully with some kind of explanation. This would be a better use of Discussion section space (see comments about the Discussion section below).

We thank the reviewer for his/her points. Lateral interfaces between FtsZ protofilaments have been extensively probed for decades and several residues have been genetically implicated in lateral interactions. However, the results from these studies were ambiguous and were often open to conflicting interpretations. Our structures and random photocrosslinking results also failed to provide cross-verification for any of these residues. The weak lateral interactions that we propose to stabilize protofilament bundles offer an explanation for the ambiguities from these studies, as we observed that both in vivo and in vitro studies suffer from insensitivity. We have significantly modified the Discussion section and have mentioned these previous reports in the Discussion section.

3) It was surprising to see no mention whatsoever of ZipA and Zap proteins that are known, in some cases quite clearly, to promote lateral interactions between pfs (or at least crosslinking), and their potential roles (see also below). Furthermore, I don't think the model in Figure 8 is all that helpful in part because it does not consider the roles of these proteins in higher order assembly of FtsZ. Perhaps some of the leucine substitution mutants functioned normally for cell division is because Zap/ZipA can compensate for an intrinsic defect in lateral interactions between pfs.

We have added discussion of the roles of ZipA and the Zap proteins to the Discussion section. Moreover, we have eliminated Figure 8 and have modified the Discussion section extensively in line with the reviewer’s points.

4) It was hard to follow the different mutants, despite having the alignment figure. Could all the mutants (E. coli residue numbers) be shown on the crystal structure of the pfs? Along the same lines, it should be made more clear up front that the crosslinking studies were done with E. coli FtsZ.

We thank the reviewer for his/her suggestion. However, when we showed the Ec residue numbers together with the Mtb numbers on the crystal structure, the figure became very crowded with text. Thus, we decided to maintain the simplicity, and instead we have shown the Ec residue numbers together with the Mtb numbers in Figure 4 and Table 2, in addition to the sequence alignment in Figure 1—figure supplement 1. However, we would be happy to modify the Figure as the reviewer indicates if he/she feels strongly.

The Discussion section was disappointing for a number of reasons outlined below:1) Although concise and well written, it is superficial and somewhat speculative and refers to a vague cartoon model that does not provide much new insight into how FtsZ protofilaments might work in the cell. The potential contributions of interfaces 1 and 2 are not discussed either. Finally, the recent evidence that FtsZ pfs move by treadmilling (Yang et al., and Bisson Filho et al., 2016) and how that activity relates to pf lateral interactions was not mentioned in the text or incorporated into the model.

We appreciate the reviewer’s suggestion and have significantly modified the discussion and have significantly modified our model.

2) Electrostatic pairing is proposed as the main mechanism for lateral interactions between protofilaments, based on the crystal structure and the predominance of charged residues at the interface. This may be true, but there is very little additional supporting evidence for this claim other than a few substitutions with leucine are disruptive. It does not help the case that one of the critical residues at interface 1 of E. coli FtsZ is a serine, not an arginine as in Mtb FtsZ. For stronger proof that electrostatics are involved, the authors should at least show that charge swaps between two known interacting residues maintains lateral interactions and in vivo function.

We thank the reviewer for his/her suggestion. We have performed additional complementation studies (Figure 5) suggesting that lateral interactions are predominantly mediated by van der Waals interactions, which are sensitive to surface geometry; the charge complementarity may enhance these associations. Moreover, these results also suggest that lateral interactions between FtsZ protofilaments are much weaker on a per subunit basis in comparison with hydrophobic longitudinal interactions. We have significantly modified the text to reflect these points.

3) Can the authors rule out the possibility that the formation of some crosslinked dimers is due to interactions with another (longitudinal) subunit within the pf instead of a lateral interaction with an adjacent pf? Insertion of pBpa at the longitudinal interface residue K140 seems to allow crosslinking (Figure 3B), but is that because pBpa at residue 140 still allows function?

We can rule out the possibility that the formation of some crosslinked dimers is due to interactions with another (longitudinal) subunit within the protofilament instead of a lateral interaction with an adjacent protofilament. This conclusion is based on the following two reasons. First, crystal structures of FtsZ protofilaments have been determined for both straight (Matsui et al., 2012, Tan et al., 2012) and curved conformations (Li et al., 2013 and this study). Simply based on crystal structures, longitudinal and lateral interfaces have no overlap and are clearly distinct from each other. Second, the longitudinal interface is very sensitive to disruptive mutation (Li et al., 2013). Based on the structure, most of the pBpa mutations will be disruptive, except residue 140 (Li et al., 2013).

We can also conclude that pBpa at residue 140 allows function. We identified residue 140 via random screening. The library was first transformed into the LY928-ΔftsZ strain to screen for variants that complemented the ftsZ-null phenotype. These variants were then subjected to in vivo photocrosslinking analysis to identify pBpa variants of FtsZ that can form crosslinked dimers.

4)I found it surprising that 8 of the leucine replacements were able to complement. Perhaps single replacements have smaller effects on lateral interactions because other interactions in the interface compensate, but the explanation offered does not seem sufficient. Perhaps changes to the opposite charge would have a larger negative effect?

We appreciate the reviewer’s concern. We have performed additional complementation studies (Figure 5) suggesting that lateral interactions are predominantly mediated by van der Waals interactions, which are sensitive to surface geometry; the charge complementarity may enhance these associations. Moreover, these results also suggest that lateral interactions between FtsZ protofilaments are much weaker on a per subunit basis in comparison with hydrophobic longitudinal interactions. The inherently weak lateral interactions observed in in vivo and in vitro studies suffer from insensitivity. However, it is precisely this property that enables FtsZ protofilaments to self-organize into a dynamic Z-ring. We have significantly modified the text to reflect these points.

5) There is no mention anywhere in the manuscript (or model in Figure 8) of the role of ZipA or Zap proteins in promoting lateral interactions between FtsZ protofilaments. The nice in vivo crosslinking results may result in part from the action of these and other FtsZ-bundling proteins and not solely from intrinsic ability of FtsZ protofilaments to interact laterally. This should be explored in the Discussion section.

We have added mention of the role of ZipA or Zap proteins in promoting lateral interactions between FtsZ protofilaments (Discussion section) and have further discussed their potential role (Discussion section).

6) It is great that the crosslinking strategy and data were internally consistent and were consistent with the functional results. However, given that ZipA and Zap proteins (among others) may influence lateral interactions between FtsZ protofilaments in vivo, the argument that the interfaces found here are important for intrinsic lateral interactions would be strengthened by testing the ability of the purified FtsZ mutants to bundle in vitro. I suggest choosing a couple of FtsZ mutants that fail to crosslink in vivo and subjecting the purified proteins to the in vitro bundling conditions used in Figure 1A-B; the prediction is that they would fail to form bundles. They already have purified K121L and D304L that were shown in Figure 6B to form protofilaments under non-bundling conditions (i.e. no DEAE-dextran), so this should be an easy experiment

We had performed similar experiments and found that they suffer from insensitivity, due to two reasons. First, in comparison with longitudinal interactions, the lateral interactions between FtsZ protofilaments are much weaker, and we observed that both in vivo and in vitro studies suffer from insensitivity. Second, in vitro assembly of FtsZ to form higher-order polymers requires non-physiological conditions, such as Ca^2+^ (Lowe and Amos, 2017), DEAE dextran (Erickson et al., 1996), or Ficoll 70 (Gonzalez et al., 2003). Therefore, the addition of bundling agents is likely critical for these effects.

By contrast, our in vivo photocrosslinking study is able to unambiguously distinguish true lateral interactions from artifacts.

7) Nevertheless, from the EM images in Figure 6B, it looks like K121L and D304L do form some paired filaments, which they shouldn't at all if they are defective in lateral interactions. Can the authors explain this contradiction?

We thank the reviewer for his/her question. in vitro assembly of FtsZ to form higher-order polymers requires non-physiological conditions, such as Ca^2+^ (Lowe and Amos, 2017), DEAE dextran (Erickson et al., 1996), or Ficoll 70 (Gonzalez et al., 2003). Therefore, the addition of the bundling agents is likely to be critical for the observed effects.

[Editors' note: the author responses to the re-review follow.]

Summary:This resubmission is an improvement over the original. It puts previous work about FtsZ bundling in better context, including other FtsZ-bundling factors such as Zap proteins, and clarifies some of the other specific concerns voiced by the first round of reviewers. Although previous microscopic and genetic evidence for lateral interactions between FtsZ filaments is quite strong, the main novelty of this study is that it is the first biochemical confirmation of lateral interactions at the atomic level between FtsZ protofilaments in cells. It also suggests that residues important for lateral interactions in vivo are different from those implicated in formation of sheets, ribbons, and bundles in vitro. This discrepancy can be rationalized by the presence of Zap proteins in vivo that keep FtsZ filaments close together, and by other proteins, e.g. FtsA, that were recently shown to do the opposite (i.e. keep FtsZ filaments apart) in a reconstituted system (PMID 28695917).

We appreciate the reviewers’ positive evaluation of our work and its significance. Below, we address the four major concerns raised by the reviewers point by point.

Essential revisions:1) Since that first submission over a year ago, the two Science papers on FtsZ treadmilling in E. coli and B. subtilis were published, establishing the importance of FtsZ filament directionality at the Z-ring. In addition, Yao et al., (2017) reported additional cryo-electron tomographic evidence for close associations between FtsZ filaments in cells. While Yao et al., argue further that FtsZ lateral interactions occur in vivo, that is merely one more piece of evidence for this. Rather, it is treadmilling of FtsZ filaments in E. coli cells that provides the biggest concern: How can antiparallel double filaments, which the authors here, claim are forming in cells, treadmill? Treadmilling requires filaments, whether single or in a group, to have the same directionality. Antiparallel double filaments of MreB, for instance, are not expected to treadmill, and they don't. How can the authors reconcile FtsZ's important treadmilling activity with their antiparallel filament model?In light of this, an alternative explanation for the authors' data is that there are many close interactions between FtsZ protofilaments that occur in vivo, potentially many which are transient in nature, and they have found a subset of them.

We thank the reviewer for reminding us of the additional cryo-electron tomographic study (PMID 28438890) and have cited it in our revised manuscript. However, we would like to clarify that, rather than reporting evidence for close associations between FtsZ filaments in cells, this study showed that a complete ring of FtsZ is not required for constriction in the early stages of bacterial cytokinesis.

The reviewer raised an outstanding question regarding the reconciliation of FtsZ treadmilling activity with our antiparallel filament model, and we appreciate and have considered at length the reviewer’s alternative explanation. Our current rationale is as follows: first, the mechanism of FtsZ treadmilling, while not yet fully resolved, is likely different from that of actin. Cooperative polymerization of single-stranded FtsZ protofilaments has long been observed (Chen et al., 2005,), along with conformational changes associated with FtsZ polymerization (Martin-Galiano et al., 2010; Matsui et al., 2012; Wagstaff et al., 2017). Taken together, it is reasonable to hypothesize that single-stranded FtsZ protofilaments can treadmill. Second, the lateral interactions between FtsZ protofilaments are transient in nature. Therefore, instead of mediating the formation of a stable and static higher order architecture, our data suggests that lateral interactions can induce changes in treadmilling velocity when protofilaments collide, which may be an important driving force in Z-ring self-organization.

We have included discussions of all of these points in the Discussion section.

2) A number of concerns were raised about Figure 7A, which shows in vitro assembly of FtsZ filaments by EM. The first is that there is no scale bar, which makes it hard to evaluate whether the filaments are single or double. Second, the field of view is so small that it is hard to make any conclusions about the overall FtsZ assembly state. But most importantly, it is clear that the D304L is forming some type of bundles or at least double stranded filaments. The authors responded to this concern in their rebuttal by discussing that "addition of the bundling agents is likely to be critical for the observed effects". However, no bundling agents were added for this experiment, and other FtsZ mutants with enhanced lateral interactions clearly have been reported to form at least double stranded filaments with no added bundling agents (see the Discussion section, which seems to ignore these results). This is in contrast to mostly single stranded filaments formed by wild type FtsZ under these conditions. If these mutant FtsZs were truly defective in lateral interactions, they should form only single stranded filaments, even perhaps in the presence of bundling agents. It is difficult to reconcile the seemingly enhanced lateral interactions observed with D304L in EM with the defective lateral interactions in vivo; the authors need to do a better job of presenting the data and discussing it.

We thank the reviewers for pointing out these problems. We have added a scale bar and enlarged the field of view.

We apologize for the confusion regarding bundling agents. We would like to clarify that the results with D304L and K121L were used as evidence that these mutants are able to polymerize into protofilaments, and we would also like to clarify that bundling agents (0.6 mg/ml DEAE-Dextran) were indeed added for the experiment. Under this condition, all FtsZ mutants except A181E formed bundles or double strands in vitro. We also observed that the bundling width is related to the concentration of DEAE-Dextran (data not shown), which indicated that bundling is connected to the addition of bundling agent. Since the DEAE-Dextran might enhance non-specific interactions between protofilaments, we were not surprised that the laterally disruptive mutants (D304L and K121L) could still form bundles in vitro. We considered the in vitro assembly states not to be convincing evidence for in vivo lateral interactions, for instance because the lateral interactions are predominantly mediated by van der Waals interactions, which are sensitive to buffer conditions. We have clarified in the revised manuscript that correlation between bundling in vitro and in vivo is complicated by these concerns.

3) The failure of the D304L and K121L mutant FtsZs to incorporate into the Z-ring described in Figure 7 is backed by some reasonably convincing theoretical data, but still is not explained well enough for general readers. The Materials and methods section does not adequately describe the procedure. In subsection “Fluorescence microscopy”, there is no indication of any inducer added, but yet in subsection “Lateral interfaces are involved in Z-ring assembly” it is mentioned that an inducer was used. How do the levels of the mutant FtsZs increase? The legend to Figure 7 needs to provide more detail about the three proteins being expressed and how they are expressed, instead of making conclusions from the data.

We apologize for the confusion; we have rewritten this section and the methods to explain much better, especially for general readers. We would like to clarify that no inducer was used in the experiments. The expression level of total FtsZ was maintained at an appropriate level for cell growth. We used different promoters to change the ratio of wild type FtsZ to mutant FtsZ.

4) The critical experiments are in Figure 7 where the authors test assembly of 3 of the FtsZ mutants in vitro and incorporation into the Z ring in vivo (2 mutants affected for crosslinking and one that is not). Panel A shows the assembly in vitro. Part of this is to show the mutant can assemble. However, the morphology of the polymers show that they appear to bundle as well as the wild type (looks like this from the dimensions of the polymers). This is of some concern but as said above the correlation between in vitro bundling and what goes on in vivo is uncertain. In panel C the authors show that the mutant gets incorporated in the Z ring when ectopically expressed at a low level. In panel E the authors do a rather complicated experiment. The idea is to see if the mutant protein gets incorporated into the ring if expressed at a higher level. The results are not so convincing, in part because this is a negative result (no incorporation) rather than a positive. The idea is that if the mutant protein copolymerizes with the WT protein there will be a mixture of filaments with variable amount of WT and mutant subunits. The argument is that if a filament has a sufficient number of mutant subunits it may be defective for lateral interaction and not be incorporated. However, there still has to be enough filaments with sufficient proportion of WT subunits so that a Z ring is formed. I find it hard to believe that adding a mutant to a level that it would not get incorporated would not cause some trouble for cell division. Division is sensitive to the level of FtsZ (30% drop blocks division) and some of the WT FtsZ has to be trapped in filaments that are nonfunctional (can't get incorporated into the Z ring) it should cause trouble. The addition of the mutant FtsZ will titrate some of the WT FtsZ away interfering with cell division. From the images shown there does not appear to not be problem as the cells appear shorter than in panel C.Two things were suggested to make this more convincing. In subsection “Lateral interfaces are involved in Z-ring assembly” (and in Figure 7E) the authors talk about a pool of intermediate size (referring to the laterally disruptive FtsZs). It follows from this that a pool of larger size should be dominant negative. This would be a simple and easy test. If the mutants behave as the authors think then overexpression of the mutants in wild type cells should destroy the Z ring whereas the control mutant should not. This should be easy to see. Also, induction of the nonaffected mutant as a control is important since overexpression of FtsZ will cause filamentation at some point. Nevertheless, the lateral mutants should be much more toxic. The authors could just see the cells filament when the mutant protein is induced. They could use a strain with ZapA-GFP and see the ring disappear when the mutant is induced. A second experiment that should be done is to express the mutants in a strain where WT ftsZ is depleted and see what structures the mutants make. Do they form a ring on their own? This could be assessed by immunofluorescence microscopy.

We thank the reviewers for these suggestions. We agree that it would be convincing if lateral mutants were to be more toxic when over-expressed in wildtype cells as suggested. However, as the reviewer mentioned, since cell division is sensitive to FtsZ levels, it would thus be difficult to distinguish the toxicity between lateral mutants and wild-type FtsZ.

Nevertheless, our complementation studies showed that lateral mutants are lethal in a strain in which wild-type FtsZ is depleted. We co-expressed fluorescent protein-tagged mutant FtsZ in wild type cells, and carefully maintained the total level of FtsZ in a range appropriate for cell division. We would like to clarify that in panel E and C, cells grew normally and formed functional Z-rings. Since we labeled the plasmid-carried mutant FtsZ with mNeonGreen, the mid-cell fluorescent signal is representative of the fraction of mutant FtsZ incorporated into the Z-ring (Figure 7—figure supplement 2). When copolymerized, the protofilaments tend to have different numbers of laterally disruptive subunits and protofilaments with fewer laterally defective subunits are more likely to get incorporated into the Z-ring. In panel E, the low fluorescence intensity suggests that more laterally subunits are more likely to remain in the cytoplasm rather than be incorporated into the Z-ring.